# MMClima: A Framework for Multimodal Climate Science Data and Evaluation

## Abstract

Climate change research increasingly requires AI systems that can operate across multiple modalities, including natural language, dynamic visual content, and scientific figures. Yet existing climate QA benchmarks remain limited: they include relatively small sets of questions, rely almost exclusively on text, and evaluate only a narrow range of models. As a result, they fail to reflect the multimodal and large-scale nature of climate knowledge. In this work, we introduce MMClima, a multimodal framework for climate question answering. MMClima contains over 104k expert-validated question–answer pairs spanning text, video transcriptions, and figures, alongside covering a diverse range of five core climate science domains. The dataset is constructed through automated claim extraction combined with human-in-the-loop validation to ensure both scale and reliability. Beyond serving as a dataset, MMClima provides a reusable framework for extending QA resources across modalities. Using MMClima, we evaluate state-of-the-art multimodal language models on tasks spanning factual recall, visual interpretation, and cross-modal synthesis. We further fine-tune on the textual split, yielding MMCLIMA-70B-TXT, a domain-adapted baseline that surpasses both open- and closed-source models. Finally, we release the dataset, evaluation pipeline, fine-tuned model weights, and data creation framework as open resources, establishing the first step toward standardized multimodal evaluation in climate science.

## 1 Introduction

Climate change is among the most consequential global challenges of our time. Large Language Models (LLMs) and Vision–Language Models (VLMs) encode vast general knowledge and are increasingly used to surface climate information, support analysis, and aid decision-making (Bulian et al., 2024; Li et al., 2024; Kuckreja et al., 2024; Lu et al., 2024; 2022). Yet high-stakes decisions in policy and infrastructure require numerically precise, source-grounded answers that adhere to conventions for evidence traceability and domain terminology (Mastrandrea et al., 2011). General-purpose models frequently struggle with fine-grained tasks such as subpanel localization, sign/unit fidelity, or event attribution, leading to unreliable performance in climate text and figure comprehension (Mukhopadhyay et al., 2024; Masry et al., 2025; 2022; Methani et al., 2020; Lu et al., 2024). Recent evaluations document similar reliability gaps even in advanced systems (Bulian et al., 2024).

Although several benchmarks have explored the scientific evaluation of LLMs in the climate domain, they remain fundamentally limited (Table 1). Small expert-curated datasets offer rigor but are restricted to text and only hundreds to a few thousand items, rendering them unsuitable for large-scale training or robust evaluation (Manivannan et al., 2025). Larger automatically generated datasets achieve scale but suffer from noise due to weak filtering and the lack of systematic human validation (Zhu & Tiwari, 2023). Even multimodal resources that include figures often rely on textual descriptions of figures and RAG setups, rather than requiring direct pixel-level figure reasoning (Mutalik et al., 2025). Collectively, existing efforts are either modest in size, unimodal in scope, or insufficiently validated, leaving open the need for a benchmark that is simultaneously large-scale, multimodal, and rigorously curated.

To address these limitations, we introduce **MMClima**, a large-scale multimodal dataset and data-creation pipeline for climate science QA. MMClima offers textual tasks (multiple-choice, cloze, and free-form) along with viusal figure-grounded tasks (multiple-choice, yes/no, and free-form) un-

Table 1: Overview of climate- and environment-focused QA datasets.

| Name | Year | Size | Automated | Validated | Multimodal | Source |
|------|------|------|-----------|-----------|------------|--------|
| Climate Crisis QA | 2024 | 19,241 | ✓ | ✗ | ✗ | Web + LLM |
| Pirá 2.0 | 2024 | 2,250 | ✗ | ✓ | ✗ | Scientific abstracts, UN reports |
| Climate-FEVER | 2020 | 1,535 | ✗ | ✓ | ✗ | Wikipedia |
| CPIQA | 2025 | 54,612 | ✓ | Partial | ✓ | Research papers (text + figures) |
| ELLE | 2025 | 1,130 | ✗ | ✓ | ✗ | Curated corpora |
| ClimaQA-Gold | 2025 | 566 | ✓ | ✓ | ✗ | Textbooks |
| ClimaQA-Silver | 2025 | 3,000 | ✓ | ✗ | ✗ | Textbooks |
| MMClima (**Ours**) | 2025 | 104,902 | ✓ | ✓ | ✓ | Wikipedia, YouTube, IPCC, reports |

der a unified protocol. All items are single-evidence grounded, ensuring attribution and auditability, and are designed to stress fine-grained requirements such as numeric precision, subpanel localization, and domain-specific terminology. Spanning five core domains of climate science, MMCLIMA provides over 104k expert-validated QA pairs and supports standardized zero-shot evaluation across state-of-the-art LLMs and VLMs. In addition, we release a domain-adapted baseline, MMCLIMA-70B-TXT, obtained by fine-tuning Llama 3.3 70B on the training split, which demonstrates the benefits of domain specialization. The modular pipeline is extensible, enabling continued expansion to new domains and sources.

**Contributions.**

- **Large-scale dataset.** A multimodal corpus of over 104k expert-validated QA pairs spanning five climate domains and multiple task forms.

- **Multimodal QA.** Integration of systematically validated figure-based questions alongside textual and video transcription-derived QA, enabling evaluation across modalities.

- **QA generation framework.** A modular pipeline that generates textual QA for new topics, combining verification against authoritative sources with decoupled claim extraction and synthesis to ensure accuracy and reduce bias.

- **Extensive benchmarking.** Standardized evaluation of 28 LLMs and 8 VLMs, covering both proprietary and open-source families.

- **Domain-adapted baseline.** Release of MMCLIMA-70B-TXT, a fine-tuned model that surpasses open- and closed-source baselines on textual QA.

## 2 RELATED WORK

Early efforts in climate-focused QA primarily relied on small, domain-specific corpora with limited annotation quality. Climate-FEVER (Diggelmann et al., 2020) introduced fact-verification pairs to address misinformation, while Pirá 2.0 (Pirozelli et al., 2024) focused on coastal and oceanic sciences through curated expert annotations. Although these resources provided strong validation, their scale is insufficient for training data-hungry models, and the narrow topical coverage restricts generalization beyond specific subfields.

Subsequent work emphasized automated data generation for broader coverage. Climate Crisis QA (Zhu & Tiwari, 2023) leveraged web sources and LLM-assisted filtering to create over 19k QA pairs, trading off annotation reliability for scale. Similarly, ClimaQA (Manivannan et al., 2025) provided both 566 Gold QA (validated, textbook-derived) and 3k Silver QA (automatically expanded) subsets, establishing the first graduate-level benchmark in climate science. However, the Silver split suffers from noise due to imperfect LLM generation, and both versions remain limited to textual sources. More recently, CPIQA (Mutalik et al., 2025) and ELLE (Guo et al., 2025) extended coverage to research papers and multi-topic environmental science, respectively, highlighting a trend toward multimodality and domain generalization. Nevertheless, even these benchmarks often underrepresent multimodal reasoning or lack rigorous validation at scale as shown in Table 1. In parallel, a broader literature on evaluating multimodal large language models (MLLMs) has emerged, proposing protocols and stress tests for trustworthy, efficient, and robust assessment of general vision–language ability, including *MMEvalPro*(Huang et al., 2025), audio–visual capability suites

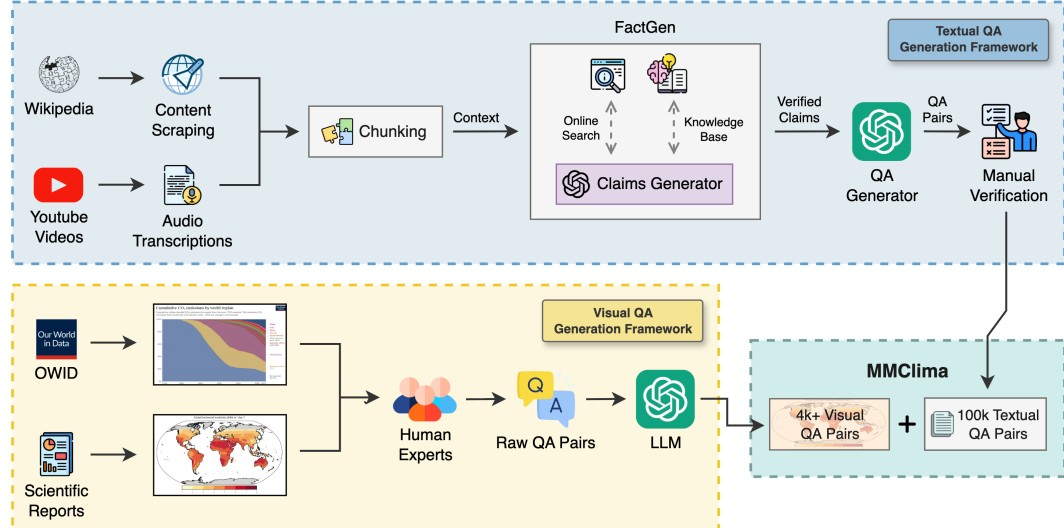

Figure 1: The MMCLIMA QA generation pipeline. Textual QA pairs are created from articles and videos via scraping, transcription, chunking, claim extraction, and automated QA synthesis with human verification. Visual QA pairs are derived from scientific figures and curated datasets, refined by human experts and LLMs. Together, these stages produce over 104k validated QA pairs, forming the first large-scale multimodal climate QA benchmark.

that examine effectiveness, efficiency, generalizability, and robustness(Zhao et al., 2025), and multimodal multi-image reasoning benchmarks (Cheng et al., 2025). While these advances calibrate evaluation for MLLMs at large, they do not target climate-science competencies (e.g., interpreting geophysical figures, units, and uncertainty statements) nor do they curate domain-grounded items with scientific provenance.

Beyond dataset construction, there is growing evidence that large language models can be effectively contextualized to improve QA quality. Techniques such as prompt-based contextual expansion (Brown et al., 2020), retrieval-augmented generation (RAG) (Lewis et al., 2020), and chain-of-thought prompting (Wei et al., 2022) have been shown to improve factual grounding and reduce hallucinations in domain-specific QA tasks. In climate science, contextualizing LLMs with structured sources (e.g., IPCC reports, Wikipedia) offers a principled way to balance coverage and reliability, yet prior datasets have not fully exploited such methods during data creation. While recent datasets offer partial coverage (e.g., *CPIQA* is multimodal but narrowly scoped; *ELLE*, *Pirá2.0* are validated but text-only; *ClimaQA-Gold* is small), Table1 highlights a gap in domain-calibrated evaluation. Climate QA demands (i) figure literacy, (ii) numeracy with scientific units and uncertainty, (iii) geo-temporal grounding, and (iv) alignment with consensus sources—criteria often unmet in general evaluations.

In summary, prior climate QA datasets underscored the value of domain-specific evaluation *but remain limited by scale, topical breadth, and modality*. For example, CLIMAQA contributed 566 validated items and a Silver split of unvalidated 3k pairs, yet its small size and text-only scope restrict generalization. MMCLIMA overcomes these limitations by providing 104k expert-validated QA pairs spanning text, video transcription, and figures across five core domains, uniting scale, diversity, and rigorous validation. This positions MMCLIMA as a substantive advance and a stronger foundation for evaluating both language and vision-language models in climate science.

## 3 MMCLIMA FRAMEWORK

Building on the limitations of existing climate QA resources, we propose **MMClima**, a large-scale multimodal framework for climate question answering with three core objectives: (i) *coverage*, by sourcing knowledge across diverse modalities such as scientific text and figures; (ii) *scale*, through automated generation of high-quality question–answer pairs; and (iii) *reliability*, ensured via system-

atic human-in-the-loop validation. The modular pipeline is extensible to new domains and modalities and yields a benchmark of over 104k validated QA pairs (Figure 1).

## 3.1 DATA SOURCES

A key design choice in MMCLIMA is to ground question generation in diverse climate materials spanning scientific and communicative contexts. To ensure breadth, we organize the corpus into five domains: *(i) Atmospheric Composition & Air Quality*, *(ii) Oceanic and Coastal Dynamics*, *(iii) Cryosphere and Glacial Systems*, *(iv) Climate-Driven Extreme Events*, and *(v) Climate Policy, Governance, and Mitigation Pathways*, each subdivided into finer-grained topics capturing processes, impacts, and policy mechanisms.

For textual sources, we rely on two complementary modalities. *Wikipedia articles* provide structured expositions of foundational climate concepts, ensuring coverage of canonical scientific knowledge (Wikimedia Foundation, 2024), while *topic-specific YouTube transcripts* capture explanatory discourse and examples reflecting how climate information is communicated to broad audiences. Together, these sources enable evaluation of models on both technical understanding and communicative reasoning.

To support multimodal evaluation, we curate visual data from two major sources. The first is the *Intergovernmental Panel on Climate Change (IPCC) Assessment Reports*, containing figures that summarize multi-decadal observations, projections, and reconstructions of key variables such as greenhouse gas emissions, global temperature, sea-level rise, cryosphere dynamics, and regional extremes (IPCC, 2021). The second is *Our World in Data (OWID)*, which provides openly licensed statistical graphics on emissions, energy systems, and climate impacts (Ritchie, Hannah and Roser, Max and Rosado, Pablo and team, 2024). These visuals are complemented by schematic diagrams (e.g., energy budgets, carbon cycles, feedback loops) and geospatial maps depicting anomalies in temperature, precipitation, and impact-driver distributions. Each visual is paired with its caption, legend, and surrounding text to preserve semantic grounding.

This combination of textual and visual resources ensures that MMCLIMA reflects both the scientific depth and communicative diversity of climate knowledge, enabling QA tasks spanning factual recall, explanatory discourse, and visually grounded reasoning (Figure 2).

## 3.2 AUTOMATED QA GENERATION

A central goal of MMClima is to construct a large-scale, high-quality textual QA benchmark grounded in climate science. To this end, we designed a multi-stage pipeline that leverages open-domain resources, large language models, and human validation. The pipeline produces 100,747 expert-validated textual QA pairs from an initial pool of 130,392 candidate claims, striking a balance between scalability and reliability.

**Source retrieval.** For each of the five domains, we programmatically retrieved relevant *Wikipedia articles* via the official API, seeded with domain-specific keywords. This yielded 5,395 candidate URLs; after deduplication and redirect resolution, 5,083 unique articles remained. We scraped and normalized their textual content, discarding stubs or pure redirects to obtain 4,938 high-quality documents. Wikipedia was selected for its breadth, structured organization, and openness for research use, complementing curated scientific resources. In parallel, we collected *educational video materials* from Youtube, obtaining 829 domain-relevant URLs. Automatic transcription extracted textual content, and videos without audio or fewer than ten words were filtered, leaving 635 usable transcripts.

**Context segmentation.** Directly providing entire articles or transcripts to LLMs is infeasible due to input length constraints. To preserve semantic coherence while remaining within typical context window limits, we employed a character-based sliding window segmentation with `max_chars=3200` and `overlap_chars=640` for long-form textual articles, and `chunk_size=500`, `overlap=50` for shorter video transcripts. The larger window was chosen for textual data to capture broader contextual dependencies, while the smaller window for video transcripts reflects their more fragmented and conversational style. This design is supported by recent evidence that smaller segments (64–128 tokens) are more effective for fact retrieval, whereas

QA Dataset Samples

| Theme | MCQs | Free Form | Cloze |
|---|---|---|---|
| Climate-Driven Extreme Events | What was the ranking of Hurricane Rick in terms of intensity among tropical cyclones worldwide in 2009? A: The most intense, B: The least intense, C: The third-most intense, D: The second-most intense **Ground Truth: D** | What areas in downtown Philadelphia are currently in the 100-year floodplain? **Answer:** Penn's Landing and the Northeast Corridor railroad tracks at 30th Street Station are currently in the 100-year floodplain. | Tropical cyclones form only over tropical regions near the equator, between the _____ and the Tropic of Capricorn. **Answer:** Tropic of Cancer |
| Climate Policy, Governance and Mitigation Pathways | What type of disease outbreaks have been associated with flood-exposed communities in Ghana? A: Malaria and tuberculosis outbreaks, B: Influenza and measles outbreaks, C: Cholera and non-cholera diarrheal disease outbreaks D: Hepatitis and typhoid outbreaks **Ground Truth: C** | How does the executive committee of the Warsaw International Mechanism guide the implementation of loss and damage functions? **Answer:** The executive committee guides the implementation through a rolling work plan that sets direction for efforts in five-year increments. | A 2022 report highlights that Switzerland's average carbon footprint is _____ tonnes of $CO_2$ per resident annually, which is markedly higher than the global average of 6 tonnes. **Answer:** 14 tonnes |

VQA Dataset Samples

| Images | MCQs | Yes or No | Open Ended |
|---|---|---|---|
|  | According to the figure, what was the approximate total $CO_2$ emissions level in 2010? A: Between 27,000-29,000 $MtCO_2$ B: Between 31,000-33,000 $MtCO_2$ C: Between 35,000-37,000 $MtCO_2$ D: Between 39,000-41,000 $MtCO_2$ **Ground Truth: B** | According to the figure, did yearly changes remain consistently positive between 2000 and 2020? **Answer: No** According to the figure, did total emissions ever exceed 37,000 $MtCO_2$ between 2000 and 2020? **Answer: Yes** | According to the figure, what was the approximate yearly change in emissions in 2020? **Answer:** −2000 MtCO |
|  | According to the figure, which warming level corresponds to a median increase in intensity of roughly 10-13% for both 10-year and 50-year events? A: 1.0 °C B: 1.5 °C C: 2.0 °C D: 3.0 °C **Ground Truth: B** | According to the figure, at 3.0 °C warming, do 10-year events reach a higher median increase in intensity than 50-year events? **Answer: No** According to the figure, do both 10-year and 50-year events show a monotonic increase in intensity as global warming levels rise? **Answer: Yes** | According to the figure, at which warming level does the median increase for 50-year events first exceed 20%? **Answer: 3.0 °C** |

Figure 2: Samples from MMCLIMA, covering textual QA (MCQ, free-form, cloze) and VQA (MCQ, yes/no, open-ended).

larger segments (512–1024 tokens) better capture complex reasoning (Bhat et al., 2025), and is consistent with best practices advocating 10–20% overlap to preserve semantic continuity (Pinecone, 2024; Weaviate, 2024; Unstructured, 2024). After preprocessing, this stage yielded 25,319 article chunks and 2,268 video chunks, details across theme is given in Appendix A.1.

**Claim extraction and deduplication.** To distill atomic, verifiable knowledge units, each chunk was processed with gpt-4.1-nano, which generated candidate factual claims (prompt details are provided in Appendix A.5.1). This stage produced 138,509 claims. To mitigate redundancy from overlapping chunks and repeated phrasing across sources, we applied semantic similarity filtering. Each claim $c_i$ was embedded using the all-MiniLM-L6-v2 encoder, yielding a representation $\mathbf{e}_i \in \mathbb{R}^d$. Pairwise similarities were computed via cosine similarity:

$$\text{sim}(c_i, c_j) = \frac{\mathbf{e}_i \cdot \mathbf{e}_j}{\|\mathbf{e}_i\| \, \|\mathbf{e}_j\|}. \tag{1}$$

We removed duplicates whenever $\text{sim}(c_i, c_j) > 0.9$, retaining only one representative per cluster of near-identical claims. This filtering reduced the pool to 122,215 unique claims, consistent with prior work demonstrating that deduplication improves dataset diversity and robustness (Pan et al., 2023).

**FactGen & QA Generation.** To ensure that extracted claims reflect verifiable climate knowledge, we introduced a *FactGen* module that cross-checks candidate claims against external resources, including targeted web search and authoritative references such as IPCC assessment reports using RAG. This verification step filters out spurious or unverifiable statements, yielding a refined pool of factually grounded claims. The validated claims are then passed to the QA generator, which transforms them into diverse question–answer formats. To mitigate model-specific phrasing biases, we decouple extraction and synthesis by using gpt-4.1-nano for claim extraction and Llama-3.3-70B-Instruct-Turbo for QA synthesis, following emerging best practices, to use different LLMs for data preparation, in synthetic dataset construction (Manivannan et al., 2025; Pan et al., 2023). The QA generator produces multiple-choice, cloze, and free-form items, capturing factual retrieval, reasoning-oriented tasks, and open-ended responses. All generated items underwent iterative human validation to guarantee scientific correctness, clarity, and topical relevance.

The resulting corpus contains **100,747 expert-validated QA pairs**, establishing MMCLIMA as one of the largest and most rigorously validated climate QA resources to date.

| Task / Source | Train Split | | | Test Split | | |
|---|---|---|---|---|---|---|
| | **Wiki** | **Video** | **Total** | **Wiki** | **Video** | **Total** |
| Cloze | 23,373 | 2,289 | 25,662 | 5,846 | 574 | 6,420 |
| Free-form | 20,950 | 1,959 | 22,909 | 5,240 | 494 | 5,734 |
| Multiple-choice | 29,727 | 2,287 | 32,014 | 7,434 | 574 | 8,008 |
| **Total** | 74,050 | 6,535 | 80,585 | 18,520 | 1,642 | 20,162 |

Table 2: Distribution of validated QA items across textual (Wiki) and video sources for cloze, free-form, and multiple-choice formats. The final MMCLIMA dataset comprises 100,747 QA pairs.

**Final corpus statistics.** The final corpus integrates validated QA pairs from both textual and video sources, covering multiple formats and domains. Table 2 provides a detailed breakdown of the distribution across sources and splits for textual QA.

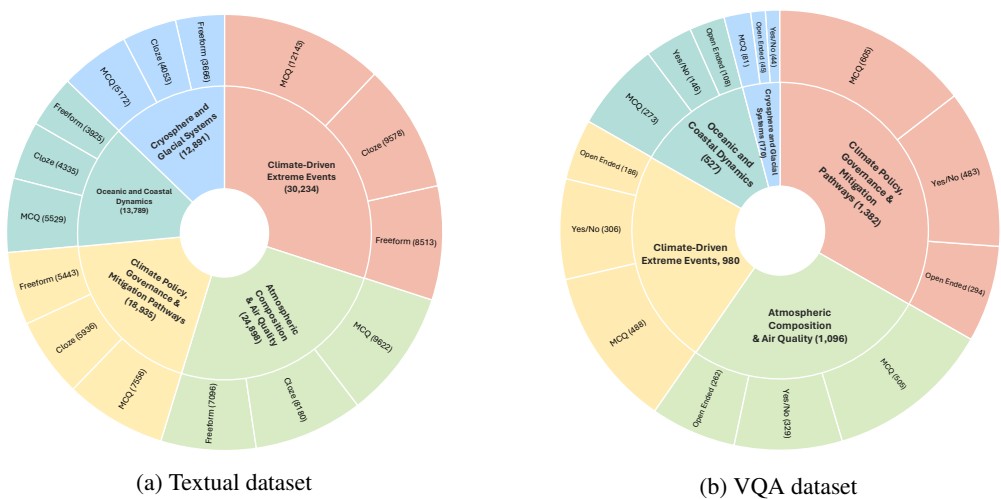

(a) Textual dataset

(b) VQA dataset

Figure 3: Sample distributions across climate topics: (a) Textual dataset and (b) VQA dataset.

### 3.3 VQA GENERATION

To construct the VQA component of MMCLIMA, we adopted a hybrid methodology combining expert-driven and automated generation. In the manual branch, four annotators examined figures from authoritative climate sources and distilled 4-7 key visual insights per figure (e.g., trends, spatial contrasts, variable dominance).

For scalability, the automated branch transformed these manually identified claims into visually grounded QA pairs using structured templates that enforced consistent constraints (e.g., balanced distractors for MCQs, concise answers with appropriate units, and JSON-formatted outputs for post-processing). This hybrid approach balances depth and quality from manual curation with coverage and scale from automation, resulting in one of the first large-scale expert-validated VQA resources in climate science.

### 3.4 HUMAN-IN-THE-LOOP VALIDATION

While automation enables large-scale generation, scientific reliability requires rigorous oversight. Following best practices in domain-specific QA (Manivannan et al., 2025), we implemented a multi-stage validation pipeline spanning both textual and visual QA. Five domain experts invested over 210 hours in iterative review, filtering ambiguous stems, unclear units, implausible distractors, and items

with answer leakage. The process combined initial screening for clarity and correctness with secondary passes verifying visual grounding, uniqueness, and consistency. Inter-annotator agreement on a stratified sample reached $\kappa = 0.83$, reflecting strong reliability.

This validation reduced noise from automated generation while preserving coverage, yielding a final corpus of **104k expert-validated QA pairs** across text, video transcripts, and figures for given themes as shown in Figure 3. By coupling scale with rigor, MMCLIMA ensures both breadth and reliability for multimodal climate reasoning.

### 3.5 EXTENSIBILITY

A central objective of MMCLIMA is extensibility. Its modular pipeline; retrieval, segmentation, claim extraction, QA generation, and validation, enables seamless integration of new domains or modalities. For instance, extending from Wikipedia text to YouTube transcripts required only an added keywords, while downstream modules remained unchanged. Similarly, incorporating IPCC figures and OurWorldInData charts into the VQA branch reused the same synthesis and validation stages. By releasing both dataset and pipeline, MMCLIMA serves not as a static benchmark but as a living resource that evolves with new domains, modalities, and sources in climate science.

## 4 TASK DESIGN & BENCHMARK SETUP

The MMCLIMA benchmark is designed to evaluate multimodal question answering (QA) in climate science across diverse task formats. All questions are drawn from the five thematic domains introduced in §3.1, ensuring coverage of atmospheric, oceanic, cryospheric, policy, and extreme-event phenomena.

**Task formulation.** Textual QA is organized into three formats: **multiple-choice (MCQ)** items testing factual recall and conceptual understanding, **cloze-style** items requiring precise short answers, and **free-form** responses enabling explanatory reasoning. Visual QA (VQA) extends this design with **MCQs**, **yes/no** items probing binary visual hypotheses, and **open-ended** responses derived directly from scientific figures. This combination balances retrieval, reasoning, and multimodal grounding.

**Evaluation metrics.** We adopt task-appropriate metrics to ensure fair evaluation across modalities. For MCQs (textual and VQA) and yes/no items (VQA), performance is measured by *accuracy*. For free-form QA, we report *BERTScore* (Zhang et al., 2020), which better captures semantic similarity than lexical overlap. For cloze-style QA, where multiple phrasings may be semantically equivalent, we define a composite metric:

$$\text{Score}_{\text{cloze}} = 0.45 \cdot \mathbf{1}\{\hat{y} = y\} + 0.10 \cdot \text{ROUGE-1}(\hat{y}, y) + 0.45 \cdot \cos(\mathbf{e}(\hat{y}), \mathbf{e}(y)), \qquad (2)$$

where $\hat{y}$ is the model prediction, $y$ is the reference, and $\mathbf{e}(\cdot)$ denotes token embeddings from a pretrained encoder. This balances exact match, surface-level similarity, and semantic alignment. For open-ended VQA, we adopt the LLM-as-a-judge paradigm (Zheng et al., 2023), using `Llama-3.3-70B-Instruct-Turbo` to assess whether model responses are correct with respect to figure content and captions.

**Benchmarking setup.** Compared to prior benchmarks like CLIMAQA (Manivannan et al., 2025), which tested only eight textual models, MMCLIMA scales substantially in scope and rigor. We evaluate 28 LLMs on textual QA and 8 VLMs on VQA, enabling the first systematic multimodal assessment in climate science. This setup establishes MMCLIMA as a foundation for measuring both domain fidelity and multimodal reasoning in high-stakes applications.

## 5 EXPERIMENTS

### 5.1 EXPERIMENTAL SETUP

We evaluate MMCLIMA on a broad spectrum of large language models (LLMs) and vision–language models (VLMs), spanning both proprietary APIs and open-source families. The bench-

mark includes (i) commercial systems, (ii) state-of-the-art open-source instruction-tuned models, and (iii) emerging lightweight models which ensure evaluation across the full capability spectrum.

All models are tested in a *zero-shot* setting with standardized prompts for each QA format (prompts in Appendix A.5.2). No in-context examples or chain-of-thought instructions are provided, isolating performance to inherent factual and reasoning ability. Inference uses greedy decoding (temperature $= 0$) with default API hyperparameters for comparability across systems, which is appropriate for our short-answer setting and consistent with prior evaluations (Lin et al., 2022; Liang et al., 2023).

In addition to general-purpose LLMs, we introduce a domain-adapted baseline, MMCLIMA-70B-TXT, obtained by fine-tuning Llama 3.3 70B on the training split of our corpus (Table 2). To efficiently adapt such a large backbone, we employ parameter-efficient fine-tuning (PEFT) with LoRA (Hu et al., 2022), training for three epochs with learning rate $1 \times 10^{-5}$, rank $r = 64$, and scaling factor $\alpha = 128$, adapting all linear layers, more details in Appendix A.7. This balances efficiency and capacity while preserving generalization, providing a strong in-domain baseline for quantifying the benefits of domain specialization over off-the-shelf models. To further evaluate the effectiveness of the training data, we fine-tuned additional models using the same setup, with results reported in Appendix A.8.

| Model | MCQ | Cloze | Freeform | Overall Weighted |
|---|---|---|---|---|
| **Closed-source Models** | | | | |
| amazon_nova-micro-v1 | 57.18 | 26.04 | 92.28 | 58.50 |
| google_gemini-flash-1.5-8b | 62.81 | 37.69 | 88.77 | 63.09 |
| microsoft_phi-4 | 72.75 | 37.88 | 92.31 | 67.65 |
| microsoft_phi-4-reasoning-plus | **78.34** | 40.43 | 92.86 | **70.54** |
| openai_gpt-4.1-nano | 65.37 | 38.42 | **93.15** | 65.65 |
| openai_gpt-5-nano | 77.84 | **43.33** | 87.82 | 69.66 |
| x-ai_grok-3-mini | 76.06 | 40.38 | 92.61 | 69.68 |
| **Open-source Models ($\geq$20B)** | | | | |
| deepseek_deepseek-chat | 81.80 | 49.52 | 93.78 | 75.03 |
| mmclima-70b-txt **(ours)** | **89.54** | **49.72** | **95.47** | **78.24** |
| meta-llama_llama-3.1-70b-instruct | 71.73 | 45.50 | 89.13 | 68.79 |
| meta-llama_llama-3.3-70b-instruct | 80.26 | 45.56 | 88.43 | 71.42 |
| meta-llama_llama-4-scout | 82.20 | 41.54 | 86.33 | 70.02 |
| mistralai_mixtral-8x7b-instruct | 61.65 | 33.39 | 91.45 | 62.16 |
| openai_gpt-oss-20b | 70.96 | 38.84 | 88.96 | 66.25 |
| openai_gpt-oss-120b | 77.36 | 40.14 | 89.67 | 69.06 |
| qwen_qwen-2.5-72b-instruct | 73.98 | 42.14 | 93.58 | 69.90 |
| qwen_qwen3-30b-a3b | 71.32 | 26.01 | 90.19 | 62.51 |
| **Open-source Models ($<$20B)** | | | | |
| google_gemma-2-9b-it | 54.41 | **38.57** | 89.75 | 60.91 |
| google_gemma-3-4b-it | 65.22 | 30.55 | 91.86 | 62.54 |
| google_gemma-3-12b-it | 61.88 | 35.36 | 90.77 | 62.67 |
| liquid_lfm-3b | 68.07 | 28.66 | 89.26 | 61.99 |
| liquid_lfm-7b | 69.18 | 30.99 | 91.16 | 63.78 |
| meta-llama_llama-3.1-8b-instruct | 49.19 | 33.35 | 89.45 | 57.33 |
| meta-llama_llama-3.2-3b-instruct | 54.65 | 28.08 | 85.28 | 56.00 |
| mistralai_mistral-7b-instruct | 34.03 | 32.53 | **91.98** | 52.85 |
| nvidia_nemotron-nano-9b-v2 | **73.40** | 31.69 | 90.86 | **65.32** |
| qwen_qwen-2.5-7b-instruct | 59.46 | 32.70 | 91.96 | 61.37 |
| tencent_hunyuan-a13b-instruct | 68.68 | 32.08 | 89.30 | 63.35 |

Table 3: Performance of closed- and open-source models on the **textual QA benchmark**. Results are reported for three task types, MCQ, Cloze, and Freeform, with the rightmost column showing the weighted overall score. The best model within each source category is marked in **bold**, while the best overall score across all models is highlighted in **dark green**.

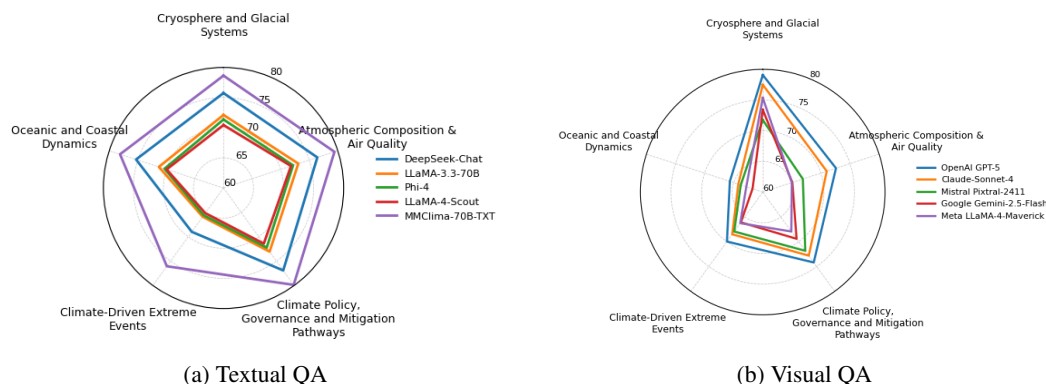

(a) Textual QA                    (b) Visual QA

Figure 4: Radar plots comparing leading models on (a) textual QA and (b) visual QA. Each axis corresponds to one of the five climate science domains introduced in §3.1.

| Model | MCQ | Task Types Yes/No | Freeform | Overall Weighted |
|---|---|---|---|---|
| *Closed-source Models* | | | | |
| anthropic_claude-sonnet-4 | 70.62 | **85.69** | 51.18 | 69.16 |
| arcee_ai-arcee-spotlight | 57.27 | 58.49 | 44.69 | 53.48 |
| google_gemini-2.5-flash | 70.03 | 79.43 | 50.61 | 66.69 |
| openai_gpt-5 | **73.21** | 83.64 | **58.99** | **71.94** |
| *Open-source Models* | | | | |
| meta-llama_llama-4-maverick | **67.44** | 82.54 | 49.22 | 66.4 |
| google_gemma-3n-E4B-it | 50.87 | 71.46 | 45.13 | 55.82 |
| mistralai_pixtral-large-2411 | 65.68 | **84.17** | **55.75** | **68.53** |
| qwen_qwen2.5-vl-72b-instruct | 66.92 | 78.19 | 52.81 | 65.97 |

Table 4: Comparison of closed- and open-source models on the proposed **VQA benchmark**. We report accuracy for multiple task formats: MCQ, Yes/No, and Freeform. The rightmost column shows a weighted aggregate across tasks. **Bold** indicates the best model within each source category (closed vs. open), while green highlights the best score overall across all models for that column.

## 5.2 RESULTS

**Textual QA.** As shown in Table 3, our fine-tuned MMCLIMA-70B-TXT sets the new state of the art with an overall score of 78.24, outperforming both closed-source frontier systems (e.g., GPT-5-Nano at 69.66, Phi-4-Reasoning-Plus at 70.54) and the strongest open-source baseline DeepSeek-Chat (75.03). This underscores both the effectiveness of MMCLIMA's training data and the value of domain adaptation, further modality based ablation given in Table 8. Free-form QA remains comparatively easy (>90 BERTScore across models), further scoring is given in Appendix A.2, while cloze QA is the hardest (<50 for all systems), serving as a stress test for exact lexical and numerical precision (detailed scoring in Appendix A.3). MCQs fall in between, clearly separating factual reasoners (e.g., LLaMA-4-Scout, 82.20) from weaker baselines.

**Visual QA.** On VQA (Table 4), GPT-5 leads overall (71.94), with balanced MCQ and free-form performance, while Claude-Sonnet-4 dominates yes/no (85.69). Among open-source models, Pixtral-Large-2411 achieves the best overall score (68.53), rivaling closed-source systems in yes/no and free-form tasks. This shows that while closed-source VLMs remain stronger on average, open-source models are increasingly competitive in targeted sub-tasks.

Radar plots in Figure 4 highlight domain-level strengths: MMCLIMA-70B-TXT achieves consistent gains across textual domains, while GPT-5 leads VQA, with Claude-Sonnet-4 strong in cryosphere and Pixtral competitive on policy visuals. Three trends emerge: (i) fine-tuned domain-specific models can surpass frontier APIs, (ii) cloze QA remains the most discriminative task, and (iii) VQA

performance is uneven, with closed-source systems dominant but open-source rivals competitive in selective domains.

# 6 CONCLUSION

We introduced MMCLIMA, the first large-scale multimodal framework for climate question answering, unifying text, video transcripts, and figure-based QA into a benchmark of over 104k expert-validated pairs. Our modular pipeline combines automated claim extraction with human-in-the-loop validation, ensuring both scale and reliability while remaining extensible to new domains and modalities. Experiments across 28 LLMs and 8 VLMs reveal clear task asymmetries; free-form QA is comparatively easy, cloze QA remains highly discriminative, and VQA continues to challenge even frontier systems. Notably, our fine-tuned MMCLIMA-70B-TXT surpasses all closed- and open-source baselines, demonstrating the value of domain adaptation on high-quality data. We will open-source the framework, dataset, and model weights, establishing MMCLIMA as both a benchmark and an evolving resource to advance multimodal climate reasoning.

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

# A APPENDIX

## A.1 CHUNK DISTRIBUTION ACROSS THEMES

This section shows the number of chunks obtained from both sources, Wikipedia and YouTube transcripts.

| Theme | Chunk Count |
|---|---|
| Atmospheric Composition & Air Quality | 8007 |
| Climate-Driven Extreme Events | 5907 |
| Climate Policy, Governance and Mitigation Pathways | 5542 |
| Cryosphere and Glacial Systems | 4242 |
| Oceanic and Coastal Dynamics | 3889 |

Table 5: Number of chunks per theme aggregated from Wikipedia and YouTube sources.

## A.2 FREEFORM DETAILED SCORING

| Model | Freeform QA Metrics | | | |
|---|---|---|---|---|
| | ROUGE-L F1 | BLEU | BERTScore | LLM.Judge |
| **Closed-source Models** | | | | |
| amazon_nova-micro-v1 | 46.60 | 36.29 | 92.28 | 36.85 |
| google_gemini-flash-1.5-8b | 32.15 | 21.52 | 88.77 | 31.86 |
| microsoft_phi-4 | 44.57 | 34.99 | 92.31 | 41.78 |
| microsoft_phi-4-reasoning-plus | 48.69 | 39.06 | 92.86 | 43.63 |
| openai_gpt-4.1-nano | **51.77** | **41.73** | **93.15** | 45.31 |
| openai_gpt-5-nano | 26.76 | 16.16 | 87.82 | 46.50 |
| x-ai_grok-3-mini | 48.97 | 38.09 | 92.61 | **50.70** |
| **Open-source Models (≥20B)** | | | | |
| deepseek_deepseek-chat | 55.84 | 45.99 | 93.78 | **53.75** |
| mmclima-70b-txt (**ours**) | **66.34** | **57.55** | **95.47** | 52.29 |
| meta-llama_llama-3.1-70b-instruct | 35.19 | 24.08 | 89.13 | 47.68 |
| meta-llama_llama-3.3-70b-instruct | 31.83 | 20.61 | 88.43 | 45.48 |
| meta-llama_llama-4-scout | 21.73 | 11.75 | 86.33 | 38.52 |
| mistralai_mixtral-8x7b-instruct | 42.44 | 31.96 | 91.45 | 40.17 |
| openai_gpt-oss-20b | 33.90 | 23.13 | 88.96 | 38.29 |
| openai_gpt-oss-120b | 37.51 | 26.96 | 89.67 | 46.79 |
| qwen_qwen-2.5-72b-instruct | 53.51 | 43.81 | 93.58 | 46.53 |
| qwen_qwen3-30b-a3b | 38.17 | 27.54 | 90.19 | 40.50 |
| **Open-source Models (<20B)** | | | | |
| google_gemma-2-9b-it | 36.54 | 26.05 | 89.75 | 36.33 |
| google_gemma-3-4b-it | 43.94 | 34.06 | 91.86 | 23.20 |
| google_gemma-3-12b-it | 40.71 | 30.83 | 90.77 | 33.06 |
| liquid_lfm-3b | 29.73 | 20.24 | 89.26 | 33.34 |
| liquid_lfm-7b | 39.56 | 29.56 | 91.16 | 38.70 |
| meta-llama_llama-3.1-8b-instruct | 35.30 | 24.91 | 89.45 | **38.86** |
| meta-llama_llama-3.2-3b-instruct | 17.44 | 8.16 | 85.28 | 32.93 |
| mistralai_mistral-7b-instruct | 45.96 | 35.27 | **91.98** | 32.89 |
| nvidia_nemotron-nano-9b-v2 | 41.18 | 31.00 | 90.86 | 37.07 |
| qwen_qwen-2.5-7b-instruct | **46.61** | **37.28** | 91.96 | 36.02 |
| tencent_hunyuan-a13b-instruct | 33.44 | 23.93 | 89.30 | 35.65 |

Table 6: Freeform QA evaluation across models. We report ROUGE-L F1, BLEU, BERTScore, and LLM-as-a-Judge accuracy (gpt 4o mini). Bold numbers denote the best within each group, and green highlights the overall best.

## A.3 CLOZE DETAILED SCORING

This section shows weighted score on cloze questions in detailed.

| Model | Cloze EM | Cloze BLEU | Cloze CosSim | Cloze Weighted |
|---|---|---|---|---|
| **Closed-source Models** | | | | |
| amazon_nova-micro-v1 | 10.89 | 13.16 | 44.06 | 26.04 |
| google_gemini-flash-1.5-8b | 20.72 | 22.67 | 58.00 | 37.69 |
| microsoft_phi-4 | 19.38 | 24.67 | 59.31 | 37.88 |
| microsoft_phi-4-reasoning-plus | 22.55 | 26.97 | 61.30 | 40.43 |
| openai_gpt-4.1-nano | 20.65 | 24.44 | 59.29 | 38.42 |
| openai_gpt-5-nano | **27.05** | **30.17** | **62.54** | **43.33** |
| x-ai_grok-3-mini | 25.63 | 28.49 | 57.77 | 40.38 |
| **Open-source Models (≥20B)** | | | | |
| deepseek_deepseek-chat | 33.88 | 37.91 | 67.73 | 49.52 |
| **mmclima-70b-txt (ours)** | **35.18** | **46.01** | **72.80** | **49.72** |
| meta-llama_llama-3.1-70b-instruct | 28.97 | 32.31 | 64.95 | 45.50 |
| meta-llama_llama-3.3-70b-instruct | 29.05 | 32.60 | 64.94 | 45.56 |
| meta-llama_llama-4-scout | 24.43 | 27.35 | 61.81 | 41.54 |
| openai_gpt-oss-20b | 20.88 | 24.05 | 60.09 | 38.84 |
| openai_gpt-oss-120b | 26.13 | 33.96 | 65.01 | 40.14 |
| qwen_qwen-2.5-72b-instruct | 25.62 | 28.42 | 61.71 | 42.14 |
| qwen_qwen3-30b-a3b | 12.69 | 13.96 | 42.00 | 26.01 |
| **Open-source Models (<20B)** | | | | |
| google_gemma-2-9b-it | **21.70** | **24.04** | **58.67** | **38.57** |
| google_gemma-3-4b-it | 13.57 | 15.06 | 50.97 | 30.55 |
| google_gemma-3-12b-it | 18.72 | 20.31 | 55.35 | 35.36 |
| liquid_lfm-3b | 10.69 | 13.11 | 50.09 | 28.66 |
| liquid_lfm-7b | 13.32 | 16.45 | 51.90 | 30.99 |
| meta-llama_llama-3.1-8b-instruct | 16.45 | 20.12 | 53.20 | 33.35 |
| meta-llama_llama-3.2-3b-instruct | 11.37 | 13.37 | 48.07 | 28.08 |
| mistralai_mixtral-8x7b-instruct | 16.10 | 20.46 | 53.55 | 33.39 |
| mistralai_mistral-7b-instruct | 13.83 | 17.99 | 54.46 | 32.53 |
| nvidia_nemotron-nano-9b-v2 | 15.97 | 18.08 | 50.44 | 31.69 |
| qwen_qwen-2.5-7b-instruct | 15.54 | 17.72 | 53.18 | 32.70 |
| qwen_qwen-turbo | 17.27 | 18.35 | 48.62 | 31.49 |
| tencent_hunyuan-a13b-instruct | 16.12 | 18.00 | 51.17 | 32.08 |

Table 7: Performance of closed- and open-source models on the **Cloze QA task**. The best model within each source category is marked in **bold**, while the best overall score across all models is highlighted in **dark green**.

## A.4 MODALITY-ABLATED TEXTUAL QA (TEXT-ONLY VS. TRANSCRIPT-ONLY)

The table explains the modality based such as text based and transcript based breakdown of the evaluation in the Table 8

## A.5 PROMPT TEMPLATES

This section provides the list of all prompts used in our pipeline to generate factual claims and to construct different types of questions from those claims.

### A.5.1 CLAIM EXTRACTION PROMPT

Once text chunks are available, this prompt is applied to extract substantial, atomic claims that are self-contained and verifiable.

| Model | MCQ | | Cloze | | Freeform | | Overall | |
|---|---|---|---|---|---|---|---|---|
| | Text | Transcr. | Text | Transcr. | Text | Transcr. | Text | Transcr. |
| **Closed-source Models** | | | | | | | | |
| amazon_nova-micro-v1 | 56.11 | 71.08 | 21.23 | 27.74 | 92.35 | 91.51 | 56.56 | 63.44 |
| openai_gpt-4.1-nano | 64.36 | 78.40 | 33.52 | 40.86 | **93.23** | **92.39** | 63.70 | 70.55 |
| google_gemini-flash-1.5-8b | 61.84 | 75.44 | 32.07 | 43.13 | 88.76 | 88.93 | 60.89 | 69.17 |
| x-ai_grok-3-mini | 75.30 | 85.89 | 36.82 | 40.07 | 92.71 | 91.62 | 68.28 | 72.53 |
| microsoft_phi-4 | 72.11 | 80.98 | 32.75 | 40.41 | 92.43 | 91.00 | 65.76 | 70.80 |
| microsoft_phi-4-reasoning-plus | **77.83** | 84.99 | 35.45 | 43.12 | 93.00 | 91.41 | **68.76** | 73.17 |
| openai_gpt-5-nano | 77.07 | **87.80** | **38.67** | **48.01** | 87.78 | 88.33 | 67.84 | **74.71** |
| qwen_qwen-turbo | 72.33 | 78.75 | 26.70 | 34.85 | 90.44 | 89.88 | 63.16 | 67.83 |
| **Open-source Models ($\geq$20B)** | | | | | | | | |
| deepseek_deepseek-chat | 81.42 | 86.76 | 45.92 | **53.60** | 93.87 | 92.83 | 73.74 | 77.73 |
| mmclima_70b_txt **(ours)** | **89.26** | **93.19** | **49.46** | 52.38 | **95.52** | **94.95** | 78.08 | 80.17 |
| openai_gpt-oss-20b | 70.15 | 83.08 | 33.24 | 43.21 | 88.92 | 89.38 | 64.10 | 71.89 |
| openai_gpt-oss-120b | 76.62 | 86.91 | 39.47 | 47.00 | 89.73 | 89.00 | 68.61 | 74.30 |
| meta-llama_llama-3.1-70b-instruct | 70.93 | 82.06 | 41.30 | 48.33 | 89.10 | 89.50 | 67.11 | 73.30 |
| meta-llama_llama-3.3-70b-instruct | 79.84 | 85.66 | 41.49 | 48.05 | 88.37 | 89.10 | 69.90 | 74.27 |
| meta-llama_llama-4-scout | 81.88 | 86.34 | 36.33 | 47.67 | 86.29 | 86.76 | 68.17 | 73.59 |
| qwen_qwen-2.5-72b-instruct | 73.31 | 82.58 | 37.20 | 46.66 | 93.67 | 92.55 | 68.06 | 73.93 |
| qwen_qwen3-30b-a3b | 70.49 | 82.02 | 22.14 | 23.47 | 90.16 | 90.51 | 60.93 | 65.33 |
| **Open-source Models ($<$20B)** | | | | | | | | |
| mistralai_mixtral-8x7b-instruct | 60.93 | 70.55 | 28.38 | 35.64 | 91.48 | 91.13 | 60.26 | 65.77 |
| meta-llama_llama-3.2-3b-instruct | 54.20 | 60.49 | 22.80 | 29.41 | 85.10 | 86.67 | 54.03 | 58.86 |
| liquid_lfm-7b | 68.52 | 77.70 | 25.62 | 32.54 | 91.27 | 89.95 | 61.80 | 66.73 |
| google_gemma-3-12b-it | 61.13 | 71.71 | 30.24 | 38.05 | 90.82 | 90.10 | 60.73 | 66.62 |
| mistralai_mistral-7b-instruct | 32.62 | 52.26 | 27.09 | 34.01 | **92.04** | 91.29 | 50.58 | 59.19 |
| qwen_qwen-2.5-7b-instruct | 58.44 | 72.60 | 27.10 | 36.47 | 91.98 | **91.81** | 59.17 | 66.96 |
| google_gemma-2-9b-it | 53.28 | 69.00 | **33.03** | **45.51** | 89.75 | 89.73 | 58.69 | 68.08 |
| tencent_hunyuan-a13b-instruct | 68.26 | 78.30 | 26.80 | 35.82 | 89.31 | 89.20 | 61.46 | 67.77 |
| meta-llama_llama-3.1-8b-instruct | 47.82 | 66.85 | 28.54 | 34.68 | 89.44 | 89.58 | 55.27 | 63.70 |
| nvidia_nemotron-nano-9b-v2 | **72.65** | **83.13** | 26.81 | 34.76 | 90.82 | 91.31 | **63.43** | **69.73** |
| liquid_lfm-3b | 67.43 | 76.31 | 23.04 | 30.32 | 89.27 | 89.08 | 59.91 | 65.24 |
| google_gemma-3-4b-it | 64.82 | 70.45 | 24.57 | 38.34 | 91.92 | 90.96 | 60.44 | 66.58 |

Table 8: Text-only vs. transcript-only ablation on the textual QA benchmark. MCQ, Cloze, and Freeform use the provided values (Text = Wikipedia; Transcr. = YouTube). *Overall* is the unweighted mean within each modality: (MCQ + Cloze + Freeform)/3. Bold marks the highest MCQ within each source block; the global best in *Overall* is highlighted in dark green.

```
You are a precise fact extractor.
From the chunk below, identify only strong claims that meet ALL of
these rules:
1.  Standalone → The claim must be understandable by itself without
extra context.
2.  Verifiable → The claim must be fact-based and checkable against
reliable sources.
3.  Universally Accepted → The claim must be widely recognized in
climate science, policy, or history.
4.  Atomic → Each claim must express exactly one fact.
5.  Concrete → Prefer entities, events, dates, laws, or measurements.
[Warning] Reject vague/general statements.
Categories:  factual, conceptual, causal, policy, statistical.
Output each claim as JSON with:  claim, category, explanation, title,
url, theme, chunk_id, pageid.
Extract only a few high-quality, universally accepted claims per
chunk.
---
Title:  chunk['title']
Theme:  chunk['theme']
URL: chunk['url']
Chunk ID: chunk['chunk_id']
PageID: chunk['pageid']
Content:  chunk['content_chunk']
---
```

### A.5.2 QUESTIONS MAKING

This section describes the prompts used to transform extracted claims into multiple-choice (MCQ), cloze, and freeform questions.

Freeform QA Prompt This prompt converts each extracted claim into a standalone question–answer pair. It ensures the question is complete and factual, with a concise answer and a whitelabel of key tokens for retrieval and evaluation.

```
You are a precise Q&A generator.
You will be given a factual claim extracted from Wikipedia and a
thematic label.
Your job is to:
1.  Determine if the claim is *about climate science, climate change,
environment, sustainability, or related topics*.
- If yes → set "climate_related":  true.
- If no → set "climate_related":  false.
2.  Regardless of this flag:
- Create one clear *standalone question* derived directly from the
claim.
- The question must be complete by itself (not vague or partial).
- Provide a *short, precise answer* also derived directly from the
claim.
- The answer must be a short but complete sentence, factual, and to
the point.
3.  Add a new field "whitelabel", which is an array of *keywords that
answer the question in short way*.
- Examples:  ["Yes"], ["No"], ["1997"], ["CO2"], ["1 meter"], ["Paris
Agreement"].
- Use the most important short tokens that answer the question.
Return JSON only in this format (no commentary, no markdown fences):
{{
"climate_related":  true/false,
"question":  "...",
"answer":  "...",
"whitelabel":  ["...", "..."]
}}
Claim:  "{claim_text}"
Context:  Title = "{title}", Theme = "{theme}"
```

MCQ QA Prompt This prompt generates exam-style multiple-choice questions (MCQs) from extracted claims. Each question includes four options with only one correct answer, ensuring the item is clear, factual, and directly grounded in the claim.

```
You are a precise exam MCQ generator.
You will be given a claim extracted from Wikipedia and a thematic
label (theme).
Your job is to:
1.  Determine if the claim is *about climate science, climate change,
environment, sustainability, or related topics*.
- If yes → set "climate_related":  true.
- If no → set "climate_related":  false.
2.  Regardless of this flag, *always create one exam-style
multiple-choice question (MCQ)* based on the claim.
3.  MCQ Requirements:
- Question for MCQ must be complete and unambiguous.
- The question must test knowledge of the claim.
- Provide exactly 4 answer options labeled A, B, C, D.
- Only ONE option must be correct.
- Wrong answers must be plausible but clearly incorrect.
- The correct answer must be directly supported by the claim.
```

```
4.  Return JSON only in this exact format (no explanation, no
commentary, no markdown fences):
{{
"climate_related":  true/false,
"question":  "...",
"options":  {{
"A":  "...",
"B":  "...",
"C":  "...",
"D":  "..."
}},
"correct_answer":  "A/B/C/D"
}}
Claim:  "{claim_text}"
Context:  Title = "{title}", Theme = "{theme}"
```

Cloze Prompt This prompt converts claims into cloze-style (fill-in-the-blank) questions. A key factual element is replaced with a blank, and the correct answer is provided along with short keywords in the whitelabel field.

```
You are a precise cloze (fill-in-the-blank) question generator.
You will be given a factual claim extracted from Wikipedia and a
thematic label.
Your job is to:
1.  Determine if the claim is *about climate science, climate change,
environment, sustainability, or related topics*.
- If yes → set "climate_related":  true.
- If no → set "climate_related":  false.
2.  Regardless of this flag:
- Create a *cloze question* (fill-in-the-blank statement) derived
directly from the claim.
- Replace the most important factual value in the claim with a blank:
"__".
- The cloze must be complete and understandable by itself (not vague
or partial).
- Provide the correct *answer* that fills the blank.
- The answer must be a short but complete factual phrase (e.g.,
"1997", "Paris Agreement", "1 meter", "Yes").
- The answer must also be added in *whitelabel* as an array of
keywords.
Return JSON only in this format (no commentary, no markdown fences):
{{
"climate_related":  true/false,
"cloze":  "...  __ ...",
"answer":  "...",
"whitelabel":  ["...", "..."]
}}
Claim:  "{claim_text}"
Context:  Title = "{title}", Theme = "{theme}"
```

### A.6    ILLUSTRATIVE DATASET SAMPLES

We present a few illustrative examples from the dataset for clarity.

#### A.6.1    SAMPLE 1: CLOZE INSTANCE

This example shows a cloze instance where the metadata field stores information to backtrack the claim, including its explanation, theme, and source URL.

```
{
    "metadata": {
      "claim": "When permafrost thaws due to global warming, large
          amounts of organic material become available for
          methanogenesis and may be released as methane.",
      "category": "causal",
      "explanation": "This claim explains the causal process linking
          permafrost thaw to methane release, a widely accepted
          mechanism in climate science.",
      "title": "Arctic methane emissions",
      "url": "https://en.wikipedia.org/?curid=19480112",
      "theme": "Cryosphere and Glacial Systems",
      "chunk_id": 1,
      "pageid": 19480112
    },
    "qa": {
      "climate_related": true,
      "cloze": "When permafrost thaws due to global warming, large
          amounts of organic material become available for ____ and
          may be released as methane.",
      "answer": "methanogenesis",
      "whitelabel": [
        "methanogenesis",
        "methane"
      ]
    },
    "source": "wikipedia",
    "id": 4408
}
```

```
{
    "metadata": {
      "claim": "High levels of air pollution in urban areas increase
          the urban heat island effect by changing the radiative
          properties of the atmosphere.",
      "category": "causal",
      "explanation": "This claim links urban air pollution to an
          increase in the urban heat island effect through altered
          atmospheric radiation, supported by atmospheric science
          research.",
      "title": "Urban heat island",
      "url": "https://en.wikipedia.org/?curid=32236",
      "theme": "Climate Policy, Governance and Mitigation Pathways",
      "chunk_id": 4,
      "pageid": 32236
    },
    "qa": {
      "climate_related": true,
      "cloze": "High levels of air pollution in urban areas increase
          the urban heat island effect by changing the ____
          properties of the atmosphere.",
      "answer": "radiative",
      "whitelabel": [
        "radiative",
        "atmosphere",
        "urban heat island"
      ]
    },
    "source": "wikipedia",
    "id": 5699
```

### A.6.2 SAMPLE 2: FREEFORM INSTANCE

```
{
    "metadata": {
      "claim": "Diammonium phosphate is used as a fire retardant
          because it lowers combustion temperature, decreases maximum
          weight loss rates, and increases residue or char
          production.",
      "category": "policy",
      "explanation": "The fire retardant properties of DAP are
          documented and used in wildfire management.",
      "title": "Diammonium phosphate",
      "url": "https://en.wikipedia.org/?curid=1722958",
      "theme": "Atmospheric Composition & Air Quality",
      "chunk_id": 1,
      "pageid": 1722958
    },
    "qa": {
      "climate_related": true,
      "question": "Why is diammonium phosphate used as a fire
          retardant?",
      "answer": "Diammonium phosphate is used as a fire retardant
          because it lowers combustion temperature, decreases maximum
          weight loss rates, and increases residue or char
          production.",
      "whitelabel": [
        "lowers combustion temperature",
        "decreases weight loss",
        "increases char production"
      ]
    },
    "source": "wikipedia",
    "id": 7315
}
```

```
{
    "metadata": {
      "claim": "Tropical cyclones typically weaken while situated
          over a landmass because conditions are often unfavorable as
          a result of the lack of oceanic forcing.",
      "category": "causal",
      "explanation": "It is a well-established fact that tropical
          cyclones lose strength over land due to the absence of warm
          ocean water energy sources.",
      "title": "Tropical cyclone",
      "url": "https://en.wikipedia.org/?curid=8282374",
      "theme": "Climate-Driven Extreme Events",
      "chunk_id": 8,
      "pageid": 8282374
    },
    "qa": {
      "climate_related": true,
      "question": "Why do tropical cyclones typically weaken when
          they are over a landmass?",
```

```
        "answer": "Tropical cyclones typically weaken over a landmass
            because conditions are often unfavorable due to the lack of
            oceanic forcing.",
        "whitelabel": [
          "Lack of oceanic forcing",
          "Unfavorable conditions"
        ]
      },
      "source": "wikipedia",
      "id": 9407
  }
```

### A.6.3  SAMPLE 3: MCQ INSTANCE

```
  {
      "metadata": {
        "claim": "Venturi scrubbers accelerate dust-laden gases to
            speeds between 12,000 and 36,000 ft/min (60.97-182.83 m/s)
            to atomize water spray into fine droplets.",
        "category": "factual",
        "explanation": "This claim provides specific measurable speeds
            for gas acceleration in venturi scrubbers, which is a
            concrete and verifiable fact in scrubber design.",
        "title": "Wet scrubber",
        "url": "https://en.wikipedia.org/?curid=8546244",
        "theme": "Atmospheric Composition & Air Quality",
        "chunk_id": 7,
        "pageid": 8546244
      },
      "mcq": {
        "climate_related": true,
        "question": "What is the speed range at which Venturi scrubbers
            accelerate dust-laden gases to atomize water spray into
            fine droplets?",
        "options": {
          "A": "1,000 to 10,000 ft/min",
          "B": "12,000 to 36,000 ft/min",
          "C": "40,000 to 60,000 ft/min",
          "D": "80,000 to 100,000 ft/min"
        },
        "correct_answer": "B"
      },
      "source": "wikipedia",
      "id": 68748
    }
```

### A.6.4 SAMPLE 4: VQA (OPENENDED)

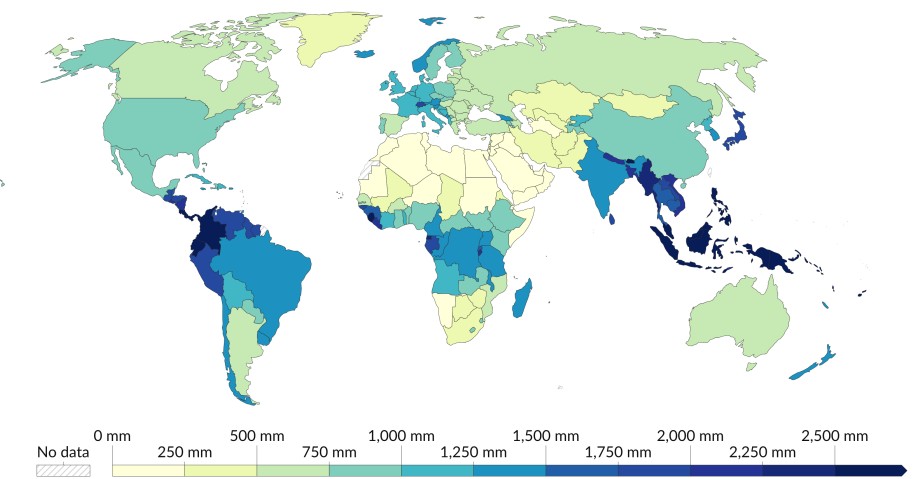

Figure 5: Precipitation patterns in South America, 2024 (`average-precipitation-per-year.png`).

```
{
    "question_type": "OpenEnded",
    "question_stem": "How does precipitation in South America vary
        across different regions in 2024?",
    "answer": "The Amazon Basin shows extremely high precipitation
        above 2,250 mm, while southern regions of South America receive
        between 750 1 ,250 mm, and western arid zones like coastal
        Peru receive less than 500 mm.",
    "explanation": "The map highlights strong contrasts within South
        America, with deep blue in the Amazon and light green to yellow
        along arid western coasts.",
    "source_topic": "Climate-Driven Extreme Events",
    "figure_name": "average-precipitation-per-year.png",
    "id": 1363
}
```

### A.6.5   SAMPLE 5: VQA (YES/NO)

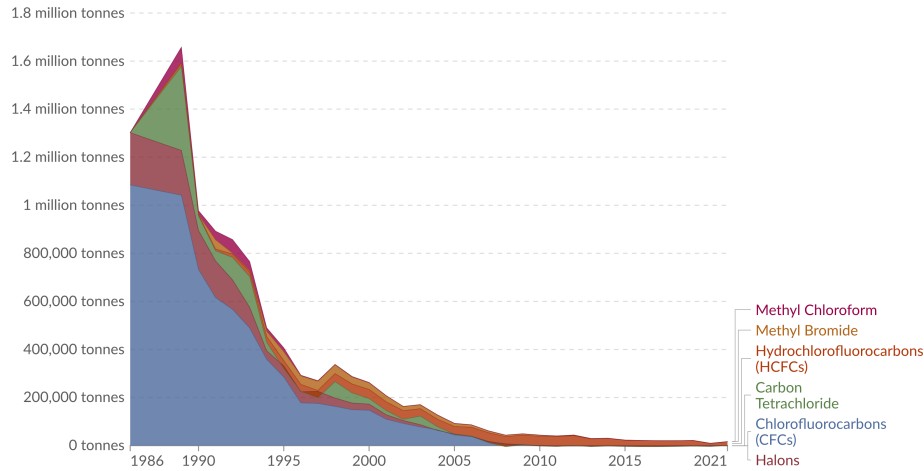

Figure 6: Consumption of ozone-depleting substances (`ozone-depleting-substance-consumption.png`).

```
{
  "question_type": "YesNo",
  "question_stem": "Did the consumption of ozone-depleting substances
      decrease significantly after the 1990s?",
  "answer": "Yes",
  "explanation": "The chart clearly shows a significant decline in
      the consumption of ozone-depleting substances after the 1990s,
      particularly after the implementation of international
      agreements such as the Montreal Protocol.",
  "source_topic": "Climate Policy, Governance, and Mitigation
      Pathways",
  "figure_name": "ozone-depleting-substance-consumption.png",
  "id": 3108
}
```

### A.6.6 SAMPLE 6: VQA (MCQ)

**Heat content in the top 700 meters of the world's oceans**

Ocean heat content is measured relative to the 1971–2000 average, which is set at zero for reference. It is measured in $10^{22}$ joules. For reference, $10^{22}$ joules are equal to approximately 17 times the amount of energy used globally every year.

Data source: EPA based on various sources (2021); NOAA National Centers for Environmental Information - Heat Content Basin Time Series (2025)

Note: Heat content is shown for four sources: Commonwealth Scientific and Industrial Research Organisation (CSIRO); Institute of Atmospheric Physics (IAP); National Oceanic and Atmospheric Administration (NOAA); and Meteorological Research Institute (MRI).

OurWorldinData.org/climate-change | CC BY

Figure 7: Upper 700m ocean heat content, 1955–2024 (`ocean-heat-content-upper.png`).

```
{
  "question_type": "MCQ",
  "question_stem": "Which organizations data shows the highest
      increase in ocean heat content in the top 700 meters from 1955
      to 2024?",
  "options": {
    "A": "NOAA",
    "B": "IAP",
    "C": "MRI/JMA",
    "D": "CSIRO"
  },
  "correct_answer": "A",
  "explanation": "The NOAA data shows the highest and steepest
      increase in ocean heat content, especially after the 1990s, as
      represented by the orange line.",
  "source_topic": "Climate-Driven Extreme Events",
  "figure_name": "ocean-heat-content-upper.png",
  "id": 3055
}
```

### A.7 FINE-TUNING DETAILS FOR MMCLIMA-70B-TXT

To establish a domain-adapted baseline, we fine-tuned Llama3.370B using the textual training split of MMCLIMA. While the main paper (Section 5.1) outlines the high-level setup, here we provide full training details for reproducibility.

**Data splits.** We used 66% of the dataset for training and reserved 33% for validation, ensuring representative coverage across all domains and QA formats.

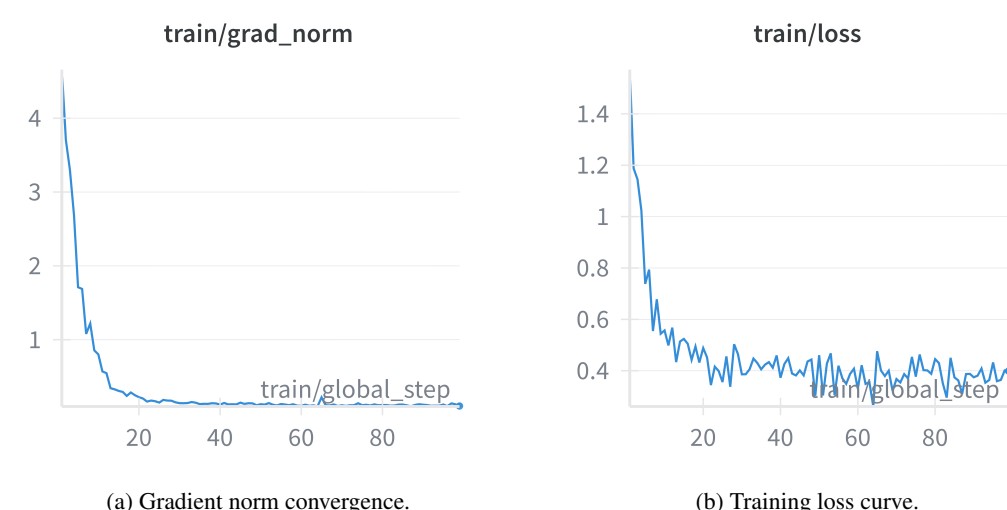

(a) Gradient norm convergence.        (b) Training loss curve.

Figure 8: Fine-tuning dynamics of MMCLIMA-70B-TXT. (a) Gradient norm stabilizes quickly, indicating well-conditioned optimization. (b) Training loss decreases sharply before plateauing near 0.4, suggesting effective domain adaptation.

**Training configuration.** The model was trained for **3 epochs** with a batch size of **8** and **1 checkpoint** saved per epoch. We performed **3 evaluation passes** at uniform intervals. Optimization followed a learning rate of $1 \times 10^{-5}$ with a linear scheduler, no warmup, weight decay set to 0, and gradient clipping at a maximum norm of 1.

**Parameter-efficient tuning.** Following LoRA (Hu et al., 2022), we set rank $r = 64$, scaling factor $\alpha = 128$, and applied adaptation to *all linear modules*. This configuration balances computational efficiency and adaptation strength while preserving generalization capacity of the base model.

**Monitoring.** Figure 8a shows gradient norm convergence, which stabilized quickly after the initial steps, while Figure 8b illustrates the steady decline in training loss, reaching $\sim$0.4 with low variance across epochs. These curves confirm stable training dynamics under our configuration.

**Outcome.** The resulting MMCLIMA-70B-TXT establishes a strong domain-adapted baseline, consistently surpassing both proprietary and open-source LLMs on textual QA (see main paper, Table 3).

### A.8 FINE-TUNING ABLATIONS

To probe the source of gains beyond the MMCLIMA-70B-TXT baseline, we conduct two complementary ablations. First, we vary supervision by stratified downsampling (70–80% of the training split), finding that even partial supervision yields consistent improvements across formats. Second, we apply the same fine-tuning recipe as Section A.7 to additional base models, *Qwen 2.5 7B Instruct* and *GPT OSS 20B*, to assess cross-model generalization.

**Setup.** Unless otherwise noted, fine-tuning settings (optimizer, scheduler, epochs, batch size, gradient clipping, and LoRA configuration) follow Section A.7. We evaluate under two input regimes per metric: *Text* (Wikipedia; text-only) and *Transcr.* (YouTube; transcript-only), reporting MCQ accuracy, Cloze (weighted), and Freeform (BERTScore).

**Results.** Table 9 shows sizeable post-FT gains for both models and both modalities. For Qwen 2.5 7B, MCQ improves by +25.53 (Text) and +14.16 (Transcr.), Cloze by +8.82 / +9.83, and Freeform by +3.09 / +2.78. For GPT OSS 20B, MCQ rises by +12.39 / +6.33, Cloze by +8.05 / +5.43, and Freeform by +5.76 / +5.27. These consistent boosts across architectures and data scales

| Model | MCQ | | Cloze | | Freeform | |
|---|---|---|---|---|---|---|
| | Text | Transcr. | Text | Transcr. | Text | Transcr. |
| Qwen 2.5 7B Instruct | 58.44 | 72.60 | 27.10 | 36.47 | 91.98 | 91.81 |
| **Fine-tuned Qwen** | **83.97** | **86.76** | **35.92** | **46.30** | **95.07** | **94.59** |
| GPT OSS 20B | 70.15 | 83.08 | 33.24 | 43.21 | 88.92 | 89.38 |
| **Fine-tuned GPT OSS** | **82.54** | **89.41** | **41.29** | **48.64** | **94.68** | **94.65** |

Table 9: Fine-tuning ablation on additional backbones. Each metric is split into *Text* (Wikipedia; text-only) and *Transcr.* (YouTube; transcript-only). Values are in the 0–100 scale (higher is better). Fine-tuning settings mirror Section A.7.

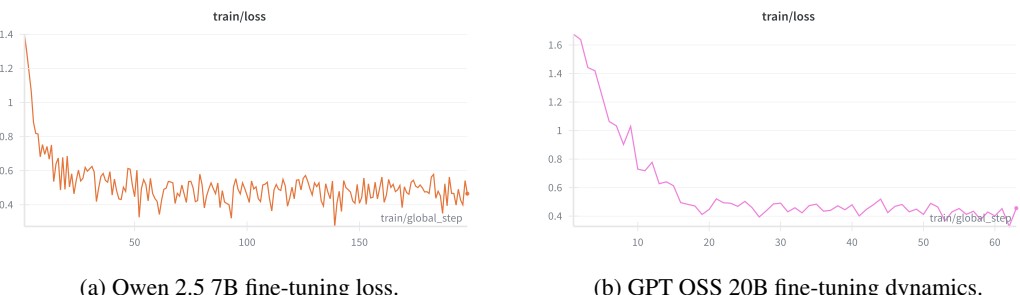

(a) Qwen 2.5 7B fine-tuning loss.        (b) GPT OSS 20B fine-tuning dynamics.

Figure 9: Fine-tuning dynamics for additional backbones. Both models exhibit stable optimization and monotonic loss reduction under the same settings as Section A.7.

indicate the improvements are not tied to a specific backbone and further validate the portability and quality of MMCLIMA supervision.

**Takeaways.** Fine-tuning on MMCLIMA reliably lifts MCQ, Cloze, and Freeform across two distinct base models and under reduced-data regimes, indicating that (i) gains are not architecture-specific and (ii) even partial supervision is effective.

### A.9 ABLATION: CONTRIBUTION OF THE VISUAL MODALITY

To probe the incremental value of the image beyond its accompanying text, we performed an ablation on a stratified random subset of 500 VQA items. We compared two settings while keeping the model, prompts, and evaluation protocol fixed: (i) *Caption/Legend Only* (the image is removed) and (ii) *Image + Caption/Legend*. Prior to this study, we manually annotated key visual trends (e.g., directions, inflections, comparative changes) to discourage questions that could be answered purely from textual context. As shown in Table 10, the *Image + Caption/Legend* setting consistently outperforms *Caption/Legend Only*, indicating substantial incremental contribution from the visual modality even when captions and legends are present.

| Model | Caption-only ↑ | | | Image+Caption ↑ | | |
|---|---|---|---|---|---|---|
| | MCQ | Yes/No | Open-ended | MCQ | Yes/No | Open-ended |
| anthropic_claude-sonnet-4 | 11.90% | 3.53% | 10.33% | 73.56% | 87.57% | 61.02% |
| google_gemini-2.5-flash | 5.24% | 2.94% | 6.67% | 71.90% | 79.41% | 60.00% |
| google_gemma-3n-e4b-it | 9.05% | 8.82% | 6.83% | 56.19% | 69.41% | 50.00% |
| meta-llama_llama-4-maverick | 6.90% | 5.88% | 7.50% | 70.00% | 85.29% | 50.00% |
| mistralai_pixtral-large-2411 | 14.29% | 7.65% | 9.17% | 70.48% | 84.12% | 66.67% |
| openai_gpt-5 | 9.05% | 2.94% | 10.00% | 76.19% | 85.29% | 66.67% |
| qwen_qwen2.5-vl-72b-instruct | 12.38% | 6.47% | 10.83% | 66.19% | 79.41% | 60.00% |

Table 10: Ablation results (subset $n=500$) comparing *Caption-only* vs. *Image+Caption* across question types (higher is better).

## A.10 Higher-tier Closed Models on a Calibrated 10% Subset

**How we built the split (for reproducibility).** Prompted by the reviewer, we adopted a calibrated subsampling procedure and fixed a **10% effective subset** for efficient evaluation of higher-tier closed models. In brief:

1. **Stratify the full benchmark** by (i) task type (MCQ, Cloze, Freeform), (ii) source/domain.

2. **Deterministic proportional sampling.** Within each stratum, sample 10% without replacement using a fixed random seed; concatenate strata to form the subset. The same locked subset is used for all models.

3. **Calibration phase (using already-run models).** Starting from the reviewer's suggestion (20%), we swept candidate rates $\{20\%, 15\%, 10\%, 5\%\}$. For each rate, and for top 10 previously evaluated models, we computed metrics on the subset and compared to full-set metrics. We selected the smallest rate that (a) kept deviations within our pre-set tolerance across metrics and (b) preserved model ranking. This yielded **10%** as our effective subset.

4. **Evaluation phase.** We then evaluated new higher-tier closed models strictly on the locked 10% subset with identical prompts and evaluation scripts.

| Model | MCQ Acc | Cloze Weighted | Free BERT |
|---|---|---|---|
| google_gemini-2.5-pro | 81.26% | 48.35% | 94.30% |
| anthropic_claude-sonnet-4 | 80.00% | 47.26% | 93.90% |
| x-ai_grok-4 | 78.81% | 44.25% | 93.40% |
| openai_gpt-5 | 83.36% | 49.94% | 94.88% |

Table 11: Results on the locked 10% calibrated subset following the reviewer's proposed strategy. "Free BERT" refers to BERTScore (F1) on free-form answers.

## A.11 The Use of Large Language Models

**LLM Usage Statement.** Large language models were employed in a limited capacity to refine the clarity and readability of the manuscript. They were not used for the conception of ideas, experimental design, analysis, or the generation of results.

## Reproducibility Statement.

To facilitate reproducibility, we describe the data pipeline in Section 3.1, including retrieval, segmentation, claim extraction, and QA generation, with validation details in Section 3.4. Experimental setup and fine-tuning configuration are provided in Section 5.1 and Appendix A.7, alongside figures tracking training dynamics. We will release the MMClima dataset, pipeline, and trained MMCLIMA-70B-TXT weights upon publication to support transparent benchmarking and future research.

