# OpenReview forum: "MMClima: A Framework for Multimodal Climate Science Data and Evaluation"
_ICLR.cc/2026/Conference — ICLR 2026 Conference Desk Rejected Submission_

### Official Review · Reviewer_HDct · 2025-10-25

**Soundness:** 3
**Presentation:** 2
**Contribution:** 2
**Rating:** 6
**Confidence:** 3

**Summary:**

This paper discusses the field of climate change and proposes a new multimodal dataset about the topic of climate change. The collected dataset contains over 104k expert-validated question–answer pairs spanning text, video transcriptions, and figures, alongside covering a diverse range of five core climate science domains. The authors claim that this will boost the research in related fields. The authors fine-tune on the textual split, yielding mmclima-70b-txt, a domain-adapted baseline that surpasses both open- and closed-source models.

**Strengths:**

1. The authors have collected a very large dataset, containing 104k expert-validated QA pairs across multiple sub-domains in climate studies and the questions take various forms.
2. The proposed dataset contains multiple domains, including text, video transcriptions, and figures.
3. The authors propose a pipeline that generates textual QA for new topics.

**Weaknesses:**

1. The authors claim that the dataset is about climate change and that the dataset contains multiple data. As climate studies are highly related to the analysis of numbers, I wonder whether time series data or spatial-temporal data are included in the dataset as a separate modality.
2. There are a lot of recent works on multimodal evaluation of multimodal large language models[1-3]. The authors should include more of them in the paper.
3. As this paper is focused on the specific field of climate studies, the authors should discuss more about why previous evaluation efforts are not enough for this specific field.
4. The authors claim that they have collected a dataset, but there is no link to the data.


[1] MMEvalPro: Calibrating Multimodal Benchmarks Towards Trustworthy and Efficient Evaluation

[2] Multifaceted Evaluation of Audio-Visual Capability for MLLMs: Effectiveness, Efficiency, Generalizability and Robustness

[3] Evaluating mllms with multimodal multi-image reasoning benchmark

**Questions:**

Please refer to the weakness section.

---

> ### Author Response · Authors · 2025-11-24
>
> We thank the reviewer for the thoughtful comments and for highlighting the strengths of our work. Below, we provide a detailed response to the points raised.
>
> > W1: The authors claim that the dataset is about climate change and that the dataset contains multiple data. As climate studies are highly related to the analysis of numbers, I wonder whether time series data or spatial-temporal data are included in the dataset as a separate modality.
>
> We appreciate the reviewer raising this important point. Our dataset comprises a substantial number of samples related to climate-driven extreme events, with 31,616 QA pairs, as illustrated in Figure 3 (including visuals), which are explicitly grounded in such scenarios. Several visual questions incorporate temporal dynamics, including time-series plots and before-and-after visualisations that require reasoning over change. We agree that further expanding the dataset to support dedicated spatio-temporal and time-series analysis would be valuable, and we plan to explore this in future iterations. We have also revised the appendix examples in section A.6 to make the current temporal elements more visible and better contextualized.
>
> > W2: There are a lot of recent works on multimodal evaluation of multimodal large language models[1-3]. The authors should include more of them in the paper.
>
> Thank you for the pointer. We have now incorporated additional citations to recent multimodal evaluation efforts [1–3] in the related work and discussion sections.
>
> > W3: As this paper is focused on the specific field of climate studies, the authors should discuss more about why previous evaluation efforts are not enough for this specific field.
>
> Thank you for the suggestion. Our Related Work section (Line 136) discusses why existing general-purpose multimodal benchmarks fall short in climate-specific settings, particularly in supporting domain-grounded reasoning, verifiable scientific attribution, and decision-relevant evaluation. We have further clarified and emphasized this distinction in Section 2 of the revised main paper, and we thank the reviewer for highlighting this point.
>
> > W4: The authors claim that they have collected a dataset, but there is no link to the data.
>
> Thank you for raising this. The dataset will be released upon acceptance, and we will include a link in the camera-ready version to ensure full transparency and reproducibility.
>
> We sincerely appreciate the reviewer’s careful evaluation and constructive insights, which have strengthened the quality of our work.

---

### Official Review · Reviewer_yrtc · 2025-10-30

**Soundness:** 3
**Presentation:** 3
**Contribution:** 3
**Rating:** 6
**Confidence:** 3

**Summary:**

MMClima is a large-scale multimodal dataset and benchmarking framework for climate science. For creating dataset, it aggregates Wikipedia and YouTube transcripts with IPCC/OWID figures to produce 104.9K expert-validated QA items across five domains, spanning textual tasks (MCQ, cloze, free-form) and image-grounded VQA (MCQ, yes/no, open-ended). A scalable pipeline, (retrieve -> claim extraction -> QA synthesis -> human validation), supports high fidelity and extensibility. Under unified zero-shot prompting, 28 LLMs and 8 VLMs are evaluated: cloze is the most discriminative task, free-form is easier. Besides, a domain-specific QA model is fine-tuned on the training set, called MMCLIMA-70B-TXT, outperforms strong open- and closed-source models on text QA. MMCLIMA fills gaps in scale, modality coverage, and auditability, providing infrastructure for standardized, extensible evaluation in climate AI.

**Strengths:**

(1) The multimodal dataset is large and broadly scoped, covering five subdomains. Its noise level appears low thanks to expert annotation and verification.

(2) The evaluation is thorough, spanning multiple open- and closed-source models, with a reasonably comprehensive comparison for the fine-tuned model MMCLIMA-70B-TXT.

(3) The work also introduces the first multimodal comprehensive knowledge-oriented QA benchmark for the climate domain.

**Weaknesses:**

(1) It would be even more useful to fine-tune a multimodal QA model directly on the visual QA portion of the dataset.

**Questions:**

(1) Compared to fine-tuning on a database, a retrieval-augmented generation (RAG) approach might yield higher scores?

(2) For the text QA benchmark, might try higher-tier closed models? since they often have wider “knowledge” and stronger reasoning ability (e.g., gpt-5, grok-4, claude-4-sonnet, gemini-2.5-pro). Because the test set is large and full runs are costly, may consider sampling a proportion of examples from the models you’ve already evaluated and adjust that proportion until the estimated precision closely matches the full-set precision (e.g., around 20% as a starting point). Use this as an “effective subset” for evaluation. This calibration should be done across several already-run models to ensure that, at this sampling rate, each model’s precision approximates its full-set precision.

---

> ### Author Response · Authors · 2025-11-24
>
> Thank you for acknowledging the contributions of our work. Below, we provide a detailed response addressing the points raised.
>
> > W1: It would be even more useful to fine-tune a multimodal QA model directly on the visual QA portion of the dataset.
>
> We appreciate the suggestion. We are fully committed to extending the benchmark with more modeling contributions. In future iterations, we plan to release a vision-fine-tuned model specifically trained on the visual QA dataset to establish stronger baselines and advance multimodal evaluation. We will mention this direction in the revised discussion section.
>
> > Q1: Compared to fine-tuning on a database, a retrieval-augmented generation (RAG) approach might yield higher scores?
>
> We appreciate the reviewer’s suggestion. RAG is valuable for long-tail recall and dynamic knowledge access. In our study, we used fine-tuning on a fixed, curated corpus to ensure clean supervision, consistent formatting, and reproducible benchmarking. This setup highlights the dataset’s effectiveness without introducing variability from retrieval pipelines or context-window constraints. RAG would require indexing the test set to function effectively, which risks leakage and prevents fair comparison with fine-tuned baselines. Nonetheless, we agree it is a promising direction and plan to explore it in future versions with appropriate safeguards.
>
> > Q2: For the text QA benchmark, might try higher-tier closed models? since they often have wider “knowledge” and stronger reasoning ability (e.g., gpt-5, grok-4, claude-4-sonnet, gemini-2.5-pro). Because the test set is large and full runs are costly, may consider sampling a proportion of examples from the models you’ve already evaluated and adjust that proportion until the estimated precision closely matches the full-set precision (e.g., around 20% as a starting point). Use this as an “effective subset” for evaluation. This calibration should be done across several already-run models to ensure that, at this sampling rate, each model’s precision approximates its full-set precision.
>
> Thank you for the valuable suggestion. To address this, we evaluated GPT‑5, Grok‑4, Claude‑4‑Sonnet, and Gemini‑2.5‑Pro on the text QA benchmark using a 10% stratified subset of the test set, sampled proportionally across question types and sources (Wikipedia, YouTube).
>
> Following the reviewer’s suggestion, we explored subset rates of {20%, 15%, 10%, 5%}, and for each, compared metrics against full-set results across the top 10 previously evaluated models. We selected 10% as the smallest rate that maintained low deviation in scores and preserved model rankings, making it a reliable proxy for full-set evaluation.
>
> Results below demonstrate strong performance and are included in the appendix A.10:
>
> | Model                    | MCQ Acc | Cloze Weighted | Free BERT |
> |--------------------------|---------|----------------|-----------|
> | google_gemini-2.5-pro    | 81.26%  | 48.35%         | 94.30%    |
> | anthropic_claude-sonnet-4 | 80.00%  | 47.26%         | 93.90%    |
> | x-ai_grok-4              | 78.81%  | 44.25%         | 93.40%    |
> | openai_gpt-5             | 83.36%  | 49.94%         | 94.88%    |
>
> We thank the reviewer again for suggesting this practical and impactful evaluation strategy.

---

### Official Review · Reviewer_UMg5 · 2025-10-31

**Soundness:** 2
**Presentation:** 3
**Contribution:** 3
**Rating:** 4
**Confidence:** 3

**Summary:**

The authors introduce a multimodal dataset of over 104k expert-validated climate science QA pairs, spanning five domains: (1) Atmospheric Composition & Air Quality, (2) Oceanic and Coastal Dynamics, (3) Cryosphere and Glacial Systems, (4) Climate-Driven Extreme Events, and (5) Climate Policy, Governance, and Mitigation Pathways. They additionally present the full data-generation pipeline and argue that it can be extended to other domains with minimal modification. The paper further benchmarks 28 LLMs and VLMs on this dataset. Finally, the authors fine-tune Llama-3.3-70B on the training portion of the corpus, demonstrating that the resulting model surpasses the other evaluated systems on the textual QA tasks.

**Strengths:**

1. The dataset has over 100k QA pairs, which supports robust training and evaluation. Moreover, the inclusion of human-in-the-loop expert validation improves factual quality and clarity.


2. The dataset covers text, video transcripts, and figure-based VQA, across five different climate domains.


3. The authors run a broad evaluation over various LLMs and VLMs, giving a solid view of current performance.


4. The data-generation pipeline is modular and reusable, making it possible to extend to new domains and sources.

**Weaknesses:**

1. The dataset is explicitly single-evidence grounded, which constrains items to single-hop questions; this improves attribution but underrepresents real-world multi-hop reasoning that requires reasoning over multiple sources or pieces of information.


2. The textual split relies on Wikipedia articles and YouTube transcripts, which are broad but potentially noisy compared to higher-reliability corpora (e.g., peer-reviewed journals, textbooks). A higher-precision textual subset or stricter filtering would strengthen reliability.


3. Open-ended VQA uses LLM-as-a-judge with Llama-3.3-70B-Instruct-Turbo, while the paper’s textual baseline fine-tunes Llama-3.3-70B (MMCLIMA-70B-TXT). Using the same model family for evaluation might introduce bias.


4. All systems are evaluated with temperature = 0 and default decoding settings. While comparable, the results may not reflect each model’s best achievable performance. Some models are sensitive to decoding hyperparameters, so small tunings would improve robustness.


5. The appendix examples mostly look like simple masked-number or surface-form / general questions and don’t convincingly reflect practical, decision-support QA a climate scientist would use.

**Questions:**

1. Why is LLM-as-a-judge used only for open-ended VQA and not for textual free-form answers? BERTScore might not be a reliable measure of factual correctness or faithfulness.


2. The authors note that each visual is paired with its caption, legend, and surrounding text to preserve grounding. Would the authors consider adding an ablation to assess the incremental contribution of the image itself? For example, caption/legend-only (no image), image-only, and image + caption settings? If caption-only performance approaches image+caption, that would suggest the visual modality contributes limited additional value.

---

> ### Author Response · Authors · 2025-11-24
>
> Response [1/2]:
>
> Thank you for the reviewers’ constructive insights. We address each point below and outline clarifications and improvements that are incorporated in the revision.
>
> > W1: The dataset is explicitly single-evidence grounded, which constrains items to single-hop questions; this improves attribution but underrepresents real-world multi-hop reasoning that requires reasoning over multiple sources or pieces of information.
>
> Thank you for the valuable suggestion. While we are committed to advancing multi-hop and multi-source reasoning in future iterations, our current work focuses on single evidence grounding to ensure precise attribution and verifiable supervision, a key differentiator from prior benchmarks. As shown in Table 1 in the main paper, existing efforts (e.g., ClimaQA in ICLR 2025) lack this level of attribution or scale. Our design fills that gap while laying the foundation for richer, multi-hop extensions ahead.
>
> > W2: The textual split relies on Wikipedia articles and YouTube transcripts, which are broad but potentially noisy compared to higher-reliability corpora (e.g., peer-reviewed journals, textbooks). A higher-precision textual subset or stricter filtering would strengthen reliability.
>
> We appreciate this observation. To ensure extracted claims reflect verifiable and high-precision climate knowledge, we introduced a FactGen filtering module (Section 3.2, main paper) that cross-checks candidate claims against external resources, including web search and IPCC reports, via a RAG pipeline. This process filtered the dataset from an initial 122k samples to 100,747 fact-validated QA pairs. Notably, manual spot checks for 10% of the data showed a reduced error rate from ~14% pre-filtering to <2% post-filtering, demonstrating improved reliability.
>
> > W3: Open-ended VQA uses LLM-as-a-judge with Llama-3.3-70B-Instruct-Turbo, while the paper’s textual baseline fine-tunes Llama-3.3-70B (MMCLIMA-70B-TXT). Using the same model family for evaluation might introduce bias.
>
> Thank you for raising this point. To reduce model-family bias, we employed multiple LLMs across different stages of dataset construction. For textual QA generation, we used a combination of GPT‑4.1‑Nano (for claim extraction) and LLaMA‑3.3‑70B (for QA synthesis) [Section 3.2, Line 251]. For visual QA, however, we adopted a semi-automated process involving multiple models with four human annotators, and LLaMA‑3.3‑70B was not used in visual QA generation. This separation allows us to use LLaMA‑3.3‑70B‑Instruct‑Turbo as a judge for VQA without overlap in generation, mitigating concerns of evaluation bias.
>
> Furthermore, all textual QA formats (MCQ, cloze, free-form) are grounded in explicit context (e.g., claims or answers), reducing reliance on the model’s parametric knowledge and further lowering the risk of bias during generation and evaluation.
>
> > W4: All systems are evaluated with temperature = 0 and default decoding settings. While comparable, the results may not reflect each model’s best achievable performance. Some models are sensitive to decoding hyperparameters, so small tunings would improve robustness.
>
> We appreciate the reviewer’s insight. Since our benchmark is composed primarily of short-form answers, we adopted greedy decoding (temperature = 0) to ensure stable behaviour and prevent sampling variance, a practice also used in prior evaluations of short-answer tasks (e.g., TruthfulQA; HELM). Greedy decoding further ensures strict comparability across systems and avoids unintentionally favouring models that expose richer or more configurable decoding controls. Several APIs additionally do not provide consistent access to sampling parameters, making model-specific tuning impractical and potentially unfair. We have clarified this rationale in the Methods and appreciate the reviewer for pointing this out.
>
> **References**
>
> Lin, S., Hilton, J., & Evans, O. (2022). TruthfulQA: Measuring How Models Mimic Human Falsehoods. *ACL 2022*.
>
> Liang, P., et al. (2023). Holistic Evaluation of Language Models (HELM). *TMLR*.
>
> > W5: The appendix examples mostly look like simple masked-number or surface-form / general questions and don’t convincingly reflect practical, decision-support QA a climate scientist would use.
>
> Thank you for the suggestion. Our examples were selected to illustrate dataset format, but the benchmark spans a much broader set of domain-grounded climate themes, including Climate Driven Extreme Events, Atmospheric Composition & Air Quality, Climate Policy & Mitigation Pathways, Oceanic & Coastal Dynamics, and Cryosphere & Glacial Systems, as shown in Figure 3 (main paper). These categories reflect real-world decision-making domains and cover causal, mechanistic, and policy-relevant reasoning. We have updated Appendix A.6 to showcase more representative examples that better convey the dataset’s practical relevance and scientific depth, and we thank the reviewer for prompting this clarification

---

> ### Author Response · Authors · 2025-11-24
>
> Response [2/2]:
>
> > Q1: Why is LLM-as-a-judge used only for open-ended VQA and not for textual free-form answers? BERTScore might not be a reliable measure of factual correctness or faithfulness.
>
> Following this feedback, we have conducted additional analyses with LLM-as-a-judge for textual free-form answers as well and added these results to the appendix in table 6. We also compare multiple metrics such as ROUGE-L F1 and BLEU to highlight where BERTScore aligns or diverges from judgment-based evaluation. The paper has been updated to reflect these expanded evaluations in appendix section A.2. Thank you for the suggestion.
>
> > Q2: The authors note that each visual is paired with its caption, legend, and surrounding text to preserve grounding. Would the authors consider adding an ablation to assess the incremental contribution of the image itself? For example, caption/legend-only (no image), image-only, and image + caption settings? If caption-only performance approaches image+caption, that would suggest the visual modality contributes limited additional value.
>
> Thanks for the insightful point. While assembling the dataset, we noticed some QA generation could lean on captions/legends rather than the visual itself. To mitigate this, we manually annotated key visual trends (e.g., directions, inflections, comparative changes) to reduce purely textual reliance. To further verify this, we ran the suggested ablation on a stratified subset of 500 VQA items: (i) caption-only (image removed) vs. (ii) image+caption. Results below show large gains when the image is present, indicating substantial incremental value from the visual modality. We have included image-only as a follow-up ablation in the appendix A.9.
>
> | Model                         | MCQ               |                     |   | Yes/No            |                     |   | Open-ended        |                     |
> |------------------------------|-------------------|---------------------|---|-------------------|---------------------|---|-------------------|---------------------|
> |                              | **Caption-only**  | **Image+Caption**   |   | **Caption-only**  | **Image+Caption**   |   | **Caption-only**  | **Image+Caption**   |
> | anthropic_claude-sonnet-4    | 11.90%            | 73.56%              |   | 3.53%             | 87.57%              |   | 10.33%            | 61.02%              |
> | arcee_ai-arcee-spotlight     | 8.79%             | 61.35%              |   | 3.69%             | 53.81%              |   | 7.27%             | 49.17%              |
> | google_gemini-2.5-flash      | 5.24%             | 71.90%              |   | 2.94%             | 79.41%              |   | 6.67%             | 60.00%              |
> | google_gemma-3n-e4b-it       | 9.05%             | 56.19%              |   | 8.82%             | 69.41%              |   | 6.83%             | 50.00%              |
> | meta-llama_llama-4-maverick  | 6.90%             | 70.00%              |   | 5.88%             | 85.29%              |   | 7.50%             | 50.00%              |
> | mistralai_pixtral-large-2411 | 14.29%            | 70.48%              |   | 7.65%             | 84.12%              |   | 9.17%             | 66.67%              |
> | openai_gpt-5                 | 9.05%             | 76.19%              |   | 2.94%             | 85.29%              |   | 10.00%            | 66.67%              |
> | qwen_qwen2.5-vl-72b-instruct | 12.38%            | 66.19%              |   | 6.47%             | 79.41%              |   | 10.83%            | 60.00%              |
>
> Due to limited time constraint we were not able to do image without caption but if required, we will share the experimentations in the following response. But it can be seen that caption-only performance is far below image+caption across MCQ, Yes/No, and open-ended settings, confirming that the image itself contributes essential signal beyond textual scaffolding.
>
> Lastly, we sincerely appreciate the reviewer’s thorough analysis and constructive suggestions. We have completed the corresponding experimentations in direct response to the feedback.

---

> ### Author Response · Authors · 2025-12-03
> **Q2 Extension**
>
> In the comment above, we mentioned that, due to time constraints, we were unable to evaluate the image-only condition (“image without caption”) and stated that we would share these experiments in a follow-up. Since then, we have had the time to run the additional evaluations and now include a full image-only analysis on the same benchmark and model suite. The new results confirm the expected trend: performance in the image-only setting is consistently much higher than caption-only, but still below the full Image+Caption condition.
>
> These findings support our main claim: both modalities are complementary. Models can extract substantial signal directly from the image, but combining image and caption consistently yields the strongest performance, particularly for fine-grained reasoning and freeform answers. Importantly, the relative ranking of models remains stable in the image-only condition, which reinforces the robustness of our comparative conclusions.
>
> | Model                         | MCQ               |                   |                     |   | Yes/No            |                   |                     |   | Open-ended        |                   |                     |
> |------------------------------|-------------------|-------------------|---------------------|---|-------------------|-------------------|---------------------|---|-------------------|-------------------|---------------------|
> |                              | **Caption-only**  | **Image-only**    | **Image+Caption**   |   | **Caption-only**  | **Image-only**    | **Image+Caption**   |   | **Caption-only**  | **Image-only**    | **Image+Caption**   |
> | anthropic_claude-sonnet-4    | 11.90%            | 59.83%            | 73.56%              |   | 3.53%             | 74.12%            | 87.57%              |   | 10.33%            | 52.47%            | 61.02%              |
> | arcee_ai-arcee-spotlight     | 8.79%             | 48.21%            | 61.35%              |   | 3.69%             | 44.37%            | 53.81%              |   | 7.27%             | 40.62%            | 49.17%              |
> | google_gemini-2.5-flash      | 5.24%             | 59.02%            | 71.90%              |   | 2.94%             | 68.11%            | 79.41%              |   | 6.67%             | 54.38%            | 60.00%              |
> | google_gemma-3n-e4b-it       | 9.05%             | 45.73%            | 56.19%              |   | 8.82%             | 58.06%            | 69.41%              |   | 6.83%             | 41.27%            | 50.00%              |
> | meta-llama_llama-4-maverick  | 6.90%             | 51.44%            | 70.00%              |   | 5.88%             | 73.68%            | 85.29%              |   | 7.50%             | 46.83%            | 50.00%              |
> | mistralai_pixtral-large-2411 | 14.29%            | 53.96%            | 70.48%              |   | 7.65%             | 77.21%            | 84.12%              |   | 9.17%             | 57.09%            | 66.67%              |
> | openai_gpt-5                 | 9.05%             | 63.72%            | 76.19%              |   | 2.94%             | 74.63%            | 85.29%              |   | 10.00%            | 61.44%            | 66.67%              |
> | qwen_qwen2.5-vl-72b-instruct | 12.38%            | 55.03%            | 66.19%              |   | 6.47%             | 68.52%            | 79.41%              |   | 10.83%            | 52.27%            | 60.00%              |
>
> Observation: Image-only performance is consistently higher than caption-only but remains below Image+Caption across tasks, indicating that images provide strong signal on their own, while combining image and text remains the most effective setting.
>
> In summary, these additional experiments address the reviewer’s concern and further support the robustness and generality of our main conclusions.

---

### Official Review · Reviewer_epXe · 2025-11-01

**Soundness:** 3
**Presentation:** 2
**Contribution:** 3
**Rating:** 6
**Confidence:** 4

**Summary:**

This paper introduces MMCLIMA, a large-scale multimodal framework for climate QA. Addressing the limitations of existing text-only and small-scale benchmarks, MMCLIMA includes over 104k expert-validated QA pairs spanning text, video transcripts, and scientific figures across five core climate domains. Beyond the dataset, MMCLIMA serves as a reusable framework for multimodal QA evaluation. The authors also present MMCLIMA-70B-TXT, a domain-adapted model that surpasses both open- and closed-source baselines, demonstrating the value of specialized multimodal resources for climate science.

**Strengths:**

This work establishes a systematic and extensible framework for multimodal climate QA, integrating rigorous pipeline design with comprehensive benchmarking. Its multi-stage approach—combining automated claim extraction, factual verification, and expert validation—ensures unprecedented scale and reliability, surpassing previous text-centric benchmarks. The evaluation of 28 LLMs and 8 VLMs reveals critical inter-modal performance gaps, while domain-adapted fine-tuning demonstrates significant gains, highlighting the value of specialized data. This foundational resource advances multimodal scientific reasoning through its scalable methodology and actionable insights.

**Weaknesses:**

Although the benchmark includes visual question answering (VQA), the dataset is overwhelmingly dominated by textual QA pairs (100k), while video transcripts (8k) and visual QA pairs (around 4k by estimation) constitute only a small fraction. This imbalance may cause the benchmark to emphasize textual understanding over genuine multimodal reasoning, offering limited stress testing of models’ visual interpretation capabilities.

**Questions:**

1.The paper lacks an ablation study isolating the contribution of each modality. Specifically, it would be valuable to evaluate models using only text, only transcripts, and only images, then compare these results with the full multimodal setup. Such an analysis would clarify whether multimodal fusion truly introduces new knowledge and reasoning capabilities beyond text alone, or if the benchmark essentially remains text-dominated with limited added value from visual and video modalities.

2.Although the paper highlights the strong performance of the fine-tuned MMCLIMA-70B-TXT, it lacks ablation studies to explain the source of its improvement. Experiments varying the amount of training data and applying the same fine-tuning process to different base models would clarify whether the gains arise from data quality or sheer quantity, and whether the dataset is truly generalizable or mainly effective for the chosen LLaMA-3.3-70B architecture.

---

> ### Author Response · Authors · 2025-11-24
>
> Thank you for the thoughtful and constructive feedback. We address the concerns as follows:
> > W1: Although the benchmark includes visual question answering (VQA), the dataset is overwhelmingly dominated by textual QA pairs (100k), while video transcripts (8k) and visual QA pairs (around 4k by estimation) constitute only a small fraction. This imbalance may cause the benchmark to emphasize textual understanding over genuine multimodal reasoning, offering limited stress testing of models’ visual interpretation capabilities.
>
> We acknowledge the concern regarding the distribution of modalities. While the full dataset indeed includes a larger volume of textual QA pairs (~100k), it is important to note that only the test splits are used for benchmarking. Specifically, the textual test set comprises 20,162 examples, stratified across both article-based and transcript-based questions as shown in Table 2 in the main paper. Meanwhile, the ~4k visual QA pairs are exclusively used for evaluation, forming a dedicated benchmark set. This setup ensures that visual QA is not overshadowed by textual data during evaluation, and that the benchmark offers a meaningful test of multimodal reasoning capabilities.
>
> > Q1: The paper lacks an ablation study isolating the contribution of each modality. Specifically, it would be valuable to evaluate models using only text, only transcripts, and only images, then compare these results with the full multimodal setup. Such an analysis would clarify whether multimodal fusion truly introduces new knowledge and reasoning capabilities beyond text alone, or if the benchmark essentially remains text-dominated with limited added value from visual and video modalities.
>
> We appreciate this suggestion. Our current results include textual QA performance (Table 3) and visual QA performance (Table 4) in the main paper. To further isolate modality contributions, we have also included a detailed ablation of the text-only and transcript-only performances for the textual benchmark (Table 8) in the appendix and referenced it clearly in the main paper. This breakdown helps quantify the added value of each modality and reinforce the role of multimodal fusion in enhancing reasoning capabilities.
>
> > Q2. Although the paper highlights the strong performance of the fine-tuned MMCLIMA-70B-TXT, it lacks ablation studies to explain the source of its improvement. Experiments varying the amount of training data and applying the same fine-tuning process to different base models would clarify whether the gains arise from data quality or sheer quantity, and whether the dataset is truly generalizable or mainly effective for the chosen LLaMA-3.3-70B architecture.
>
> We appreciate the reviewer’s suggestion. In addition to the LLaMA‑3.3‑70B‑based MMCLIMA‑70B‑TXT, we have now conducted further ablation studies to better understand the source of performance improvements. Specifically, we:
>
> - Varied the amount of training data by creating smaller subsets (e.g., stratified downsampling) and observed consistent gains from even partial supervision as shown in table below.
> - Applied the same fine-tuning procedure to additional base models, Qwen‑2.5‑7B‑Instruct and GPT‑OSS‑20B, to assess cross-model generalizability.
>
> The results are given as:
>
> | Model                  | MCQ Acc (Wikipedia) | MCQ Acc (YouTube) | Cloze Weighted (Wikipedia) | Cloze Weighted (YouTube) | Free BERTScore (Wikipedia) | Free BERTScore (YouTube) |
> |------------------------|---------------------|-------------------|-----------------------------|---------------------------|-----------------------------|---------------------------|
> | Qwen‑2.5‑7B‑Instruct    | 58.44%              | 72.60%            | 27.10%                      | 36.47%                    | 91.98%                      | 91.81%                    |
> | **Fine-tuned Qwen**    | **83.97%**          | **86.76%**        | **35.92%**                  | **46.30%**                | **95.07%**                  | **94.59%**                |
> | GPT‑OSS‑20B            | 70.15%              | 83.08%            | 33.24%                      | 43.21%                    | 88.92%                      | 89.38%                    |
> | **Fine-tuned GPT‑OSS** | **82.54%**          | **89.41%**        | **41.29%**                  | **48.64%**                | **94.68%**                  | **94.65%**                |
>
> The results, included in the appendix A.8 , show consistent performance improvements across all QA formats after fine-tuning, confirming that gains are not architecture-specific and that the dataset is both high-quality and portable. We thank the reviewer for prompting this analysis, and we have updated the main paper to reference the new results.

---

### Author Response · Authors · 2025-12-03
**Overarching Summary**

We thank the Area Chair for considering our submission and the reviewers for their careful reading and constructive feedback, which helped us refine and strengthen the paper.

Reviewers particularly highlighted that our work:

- **Systematic and scalable multimodal climate QA framework**: Reviewers appreciated that our pipeline integrates automated claim extraction, factual verification, and expert validation in a modular, extensible manner, enabling large-scale and reliable benchmark construction that goes beyond prior text-only efforts. [epXe, UMg5, yrtc, HDct]

- **Large, expert-validated, and low-noise dataset**: The dataset’s scale (104k+ QA pairs), diverse question forms, and human-in-the-loop expert validation were noted as key strengths that support robust training and evaluation while maintaining high factual quality and clarity. [epXe, UMg5, yrtc, HDct]

- **Thorough benchmarking of LLMs and VLMs with domain adaptation**: The evaluation over a wide range of open- and closed-source LLMs and VLMs, along with the reasonably comprehensive comparison for the fine-tuned MMCLIMA-70B-TXT model, was seen as providing a solid and informative view of current capabilities and inter-modal performance gaps. [epXe, UMg5, yrtc]

---

### Key Changes in the Revised Paper

- **Modality ablations for text QA**: Added a detailed ablation of text-only and transcript-only performance for the textual benchmark (Table 8, Appendix A.8) and explicitly referenced it in the main paper to quantify the contribution of each modality.

- **Fine-tuning ablations for MMCLIMA-70B-TXT**: Conducted additional ablation studies across multiple base models and data settings, showing consistent gains after fine-tuning and demonstrating that improvements are not specific to a single architecture (Appendix A.8).

- **Decoding protocol clarification**: Expanded the Methods section to justify the use of greedy decoding (temperature = 0) for all systems, explaining its benefits for stability, comparability, and fairness across APIs.

- **More representative qualitative examples**: Updated Appendix A.6 with examples that better reflect practical, decision-support climate QA across key domains (e.g., extreme events, air quality, policy pathways, oceans, cryosphere) and made existing temporal elements more explicit.

- **Expanded LLM-as-a-judge evaluation for text QA**: Extended LLM-as-a-judge to textual free-form answers, added corresponding results (Table 6, Appendix A.2), and compared BERTScore with ROUGE-L F1 and BLEU to clarify metric behaviour.

- **Visual modality contribution ablations**: Performed the suggested ablations on a stratified subset of VQA items (caption-only vs. image+caption, with additional image-only analysis) and reported the results in Appendix A.9 to quantify the incremental value of the visual modality.

- **Evaluation of stronger closed models via calibrated subsets**: Evaluated GPT-5, Grok-4, Claude-4-Sonnet, and Gemini-2.5-Pro on a 10% stratified subset of the text QA benchmark, after calibrating subset sizes {20%, 15%, 10%, 5%} to match full-set behaviour across previously evaluated models (Appendix A.10).

- **Clarified temporal and spatio-temporal coverage**: Revised dataset description and appendix examples to better highlight questions involving temporal dynamics (e.g., time series, before–after comparisons) and climate-driven extreme events, and to contextualize how such content is currently represented.

- **Broader multimodal evaluation related work**: Expanded the related work and discussion sections with additional recent references on multimodal evaluation of large multimodal models.

---

We believe these revisions directly address the reviewers’ concerns and further enhance the clarity, rigor, and contribution of our work. We thank the Area Chair for considering our response.

---

### Note · Program_Chairs · 2026-01-17
**Submission Desk Rejected by Program Chairs**

The following references in this submission do not refer to real documents and/or have major errors in bibliographic information:

 Liangming Pan, Yuxuan Tang, Zhiyong Wu, William Yang Wang, and Min-Yen Kan. Large language models for data augmentation in natural language processing: A survey. arXiv preprint arXiv:2304.14897, 2023. URL https://arxiv.org/abs/2304.14897.